# Single-atom alloy catalysts designed by first-principles calculations and artificial intelligence

Zhong-Kang Han[1,5], Debalaya Sarker[1,5], Runhai Ouyang[2,5], Aliaksei Mazheika [3✉], Yi Gao [4✉] & Sergey V. Levchenko [1✉]

Single-atom-alloy catalysts (SAACs) have recently become a frontier in catalysis research. Simultaneous optimization of reactants' facile dissociation and a balanced strength of intermediates' binding make them highly efficient catalysts for several industrially important reactions. However, discovery of new SAACs is hindered by lack of fast yet reliable prediction of catalytic properties of the large number of candidates. We address this problem by applying a compressed-sensing data-analytics approach parameterized with density-functional inputs. Besides consistently predicting efficiency of the experimentally studied SAACs, we identify more than 200 yet unreported promising candidates. Some of these candidates are more stable and efficient than the reported ones. We have also introduced a novel approach to a qualitative analysis of complex symbolic regression models based on the data-mining method subgroup discovery. Our study demonstrates the importance of data analytics for avoiding bias in catalysis design, and provides a recipe for finding best SAACs for various applications.

[1] Center for Energy Science and Technology, Skolkovo Institute of Science and Technology, Skolkovo Innovation Center, Moscow, Russia. [2] Materials Genome Institute, Shanghai University, Shanghai, P.R. China. [3] Technische Universität Berlin, BasCat—UniCat BASF JointLab, Berlin, Germany. [4] Shanghai Advanced Research Institute, Chinese Academy of Sciences, Shanghai, P.R. China. [5] These authors contributed equally: Zhong-Kang Han, Debalaya Sarker, Runhai Ouyang. ✉email: alex.mazheika@gmail.com; gaoyi@zjlab.org.cn; S.Levchenko@skoltech.ru

Recently, single-atom dispersion has been shown to dramatically reduce the usage of rare and expensive metals in heterogeneous catalysis, at the same time providing unique possibilities for tuning catalytic properties[1,2]. The pioneering work by Sykes and co-workers[2] has demonstrated that highly dilute bimetallic alloys, where single atoms of Pt-group are dispersed on the surface of an inert metal host, are highly efficient and selective in numerous catalytic reactions. These alloy catalysts are now extensively used in the hydrogenation-related reactions such as hydrogenation of $CO_2$, water–gas shift reaction, hydrogen separation, and many others[3–5]. The outstanding performance of SAACs is attributed to a balance between efficiency of $H_2$ dissociation and binding of H at the surface of metallic alloys[2,6,7].

Using desorption measurements in combination with high-resolution scanning tunneling microscopy, Kyriakou et al. have shown that isolated Pd atoms on a Cu surface can substantially reduce the energy barrier for both hydrogen uptake and subsequent desorption from the Cu metal surface[2]. Lucci and co-workers have observed that isolated Pt atoms on the Cu(111) surface exhibit stable activity and 100% selectivity for the hydrogenation of butadiene to butenes[8]. Liu et al. have investigated the fundamentals of CO adsorption on Pt/Cu SAAC using a variety of surface science and catalysis techniques. They have found that CO binds more weakly to single Pt atoms in Cu(111), compared to larger Pt ensembles or monometallic Pt. Their results demonstrate that SAACs offer a new approach to design CO-tolerant materials for industrial applications[9]. To date, Pd/Cu[10–12], Pt/Cu[7–9,13–15], Pd/Ag[12,16], Pd/Au[12], Pt/Au[17], Pt/Ni[18], Au/Ru[19], and Ni/Zn[20] SAACs have been synthesized and found to be active and selective towards different hydrogenation reactions. However, the family of experimentally synthesized SAACs for hydrogenation remains small and comparisons of their catalytic properties are scarce.

Conventional approaches to designing single-atom heterogeneous catalysts for different industrially relevant hydrogenation reactions mainly rely on trial-and-error methods. However, challenges in synthesis and in situ experimental characterization of SAACs impose limitations on these approaches. With advances in first-principles methods and computational resources, theoretical modeling opens new opportunities for rational catalyst design[6,21–48]. A general simple yet powerful approach is the creation of a large database with first-principles based inputs, followed by intelligent interrogation of the database in search of materials with the desired properties[35,48]. Significant efforts have been made in developing reliable descriptor-based models following the above general approach[6,21–35,48]. In catalysis, a descriptor is a parameter (a feature) of the catalytic material that is easy to evaluate and is correlated with a complex target property (e.g., activation energy or turnover frequency of a catalytic reaction). A notable amount of research has been devoted to searching for and using descriptors with a simple (near-linear) relation to target properties[22–30]. For example, the linear relationship between the reaction energies and the activation energies is known as the Brønsted–Evans–Polanyi relationship (BEP) in heterogeneous catalysis[29,30,45–47]. Also, the linear correlation between d-band center of a clean transition-metal surface and adsorption energies of molecules on that surface have been studied in great detail and widely applied[22–24,36,44]. In catalysis, near-linear correlations between adsorption energies of different adsorbates are referred to as scaling relations[26,28,37]. The advantages of such correlations are their simplicity and usually clear physical foundations. However, they are not exact, and there is an increasing number of studies focused on overcoming limitations imposed by the corresponding approximations[6,31–34,38–41,48]. The nonlinear and intricate relationship between the catalysts' properties and surface reactions at realistic conditions[42,43] has held back the reliable description of catalytic properties. Note that, although the stability of SAACs is of no less significance in designing a potential catalyst than their catalytic performance, it hasn't received the same level of attention.

In this work, combining first-principles calculations and compressed-sensing data-analytics methodology, we address the issues that inhibit the wider use of SAAC in different industrially important reactions. By identifying descriptors based only on properties of the host surfaces and guest single atoms, we predict the binding energies of H ($BE_H$), the dissociation energy barriers of $H_2$ molecule ($E_b$), the segregation energies (SE) of the single guest atom at different transition metal surfaces, and the segregation energies in the presence of adsorbed hydrogen ($SE_H$). The state-of-the-art compressed-sensing based approach employed here for identifying the key descriptive parameters is the recently developed SISSO (sure independence screening and sparsifying operator)[49]. SISSO enables us to identify the best low-dimensional descriptor in an immensity of offered candidates. The computational time required for our models to evaluate the catalytic properties of a SAAC is reduced by at least a factor of one thousand compared to first-principles calculations, which enables high-throughput screening of a huge number of SAAC systems.

## Results and discussion

The $BE_H$ for more than three hundred SAACs are calculated within the framework of DFT with RPBE exchange-correlation functional. This large dataset consists of $BE_H$ values at different low-index surface facets including fcc(111), fcc(110), fcc(100), hcp(0001), and bcc(110) and three stepped surface facets including fcc(211), fcc(310), and bcc(210) of SAACs with twelve transition-metal hosts (Cu, Zn, Cr, Pd, Pt, Rh, Ru, Cd, Ag, Ti, Nb, and Ta). On each TM host surface, one of the surface atoms is substituted by a guest atom to construct the SAACs. $BE_H$ for pristine surfaces (where the guest atom is the same with the host metal) are also included. H atom is placed at different non-equivalent high-symmetry sites close to the guest atom (Supplementary Fig. 1), and the $BE_H$ for the most favorable site is included in the data set. Complete information on adsorption sites and the corresponding $BE_H$ is given in Supplementary Data 1. The $BE_H$ are further validated by a comparison with previous calculations[6,21].

To better understand the variation in $BE_H$ for different guest atoms, we first investigate correlation between $BE_H$ and the d-band center of the d orbitals that are projected to the single guest atom for the alloyed systems. We find that this way of calculating d-band center provides better correlation with other properties than d-band centers for the d orbitals projected on (i) the single guest atom plus it's 1st nearest neighbor shell or (ii) the whole slab[50]. The correlation is shown in Fig. 1a (Supplementary Fig. 2) for different SAACs on Ag(110) host surface [Pt(111) host surface]. According to the d-band center theory[21,23,36,44], the closer the d-band center is to the Fermi level, the stronger the $BE_H$ should be. However, it is evident from Fig. 1a (Supplementary Fig. 2) that the expected linear correlation, as predicted by the d-band model, is broken for SAACs for H adsorption. This is due to the small size of the atomic H orbitals, leading to a relatively weak coupling between H s and the TM d-orbitals[21]. Furthermore, we check the validity of the BEP relations between the $E_b$ and the $H_2$ dissociation reaction energy for SAACs (Fig. 1b), which is commonly used to extract kinetic data for a reaction on the basis of the adsorption energies of the reactants and products[29,45–47]. As shown in Fig. 1b, the highlighted SAACs inside the blue dotted circle significantly reduce $E_b$ while reducing reaction energy only moderately. As a result, SAACs provide small reaction energy and

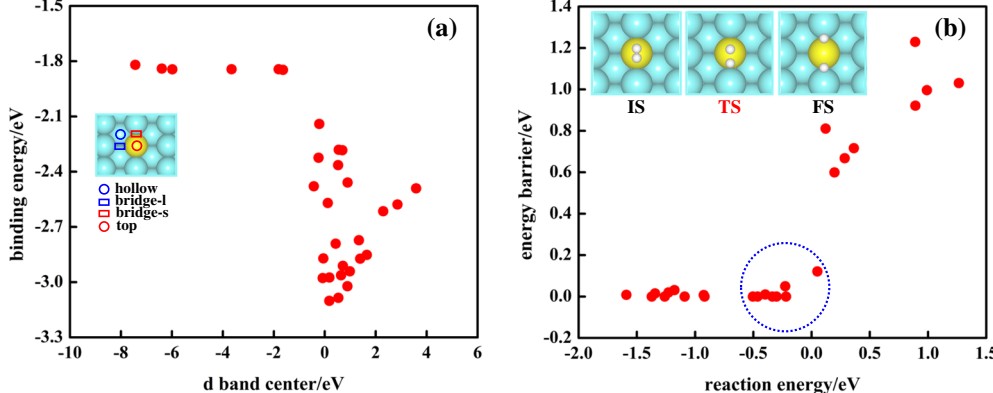

**Fig. 1 Correlation between simple descriptors and target properties.** Correlation between **a** H-atom binding energy $BE_H$ and the $d$-band center and **b** the $H_2$ dissociation energy barrier $E_b$ and the $H_2$ dissociation reaction energy for Ag(110) based SAACs. Only most stable adsorption sites are included (the hollow site for all systems on this plot). The SAACs inside the blue dotted circle in **b** significantly reduce $E_b$ while reducing reaction energy only moderately.

## Table 1 Primary features used for the descriptor construction.

| System | Class | Name | Abbreviation |
|---|---|---|---|
| Host | Atomic | Energy of the highest-occupied Kohn–Sham level | H* |
| | | Energy of the lowest-unoccupied Kohn–Sham level | L* |
| | | Electron affinity (Atomic radius) | EA*(R*)[a] |
| | | Ionization potential | IP* |
| | | Binding energy of H with single host metal atom (Binding energy of host metal dimers) | EH*(EB*)[a] |
| | | Binding distance of H with single host metal atom (Binding distance of host metal dimer) | dH*(dd*)[a] |
| | Bulk | Cohesive energy | EC* |
| | | $d$-band center | DC* |
| | Surface[b] | $d$-band center of the top surface layer | DT* |
| | | $d$-band center of the subsurface layer | DS* |
| | | Slab Fermi level | F* |
| Guest atom | Atomic | Energy of the highest-occupied Kohn–Sham level | H |
| | | Energy of the lowest-unoccupied Kohn–Sham level | L |
| | | Electron affinity (Atomic radius) | EA(R)[a] |
| | | Ionization potential | IP |
| | | Binding energy of H with single guest metal atom (Binding energy of guest metal dimers) | EH(EB)[a] |
| | | Binding distance of H with single guest metal atom (Binding distance of guest metal dimers) | dH(dd)[a] |
| | Bulk | Cohesive energy | EC |
| | | $d$-band center | DC |

[a]The feature in parentheses is used for the model of segregation energy (SE), while the feature outside parentheses is used for the models of H binding energy ($BE_H$) and $H_2$ dissociation energy barrier ($E_b$).
[b]The host metal-based features are marked by *. The surface-based primary features were calculated using the slab unit cell consisting of one atom per atomic layer.

low activation energy barrier, which leads to breaking BEP relations and thus optimized catalytic performance. The BEP relations are also found to be broken for other reactions catalyzed by SAACs[6].

Thus, the standard simple correlations (from $d$-band center theory and the BEP relations) fail for H adsorption on SAACs. Moreover, the calculation of the $d$-band center for each SAAC is highly computationally demanding, considering the very large number of candidates. These facts emphasize the necessity to find new accurate, but low-cost descriptors for computational screening of SAACs. In the SISSO method, a huge pool of more than 10 billion candidate features is first constructed iteratively by combining 19 low-cost primary features listed in Table 1 using a set of mathematical operators. A compressed-sensing based procedure is used to select one or more most relevant candidate features and construct a linear model of the target property (see Supplementary Methods for details on the SISSO procedure).

Note that the three primary surface features are properties of the pure host surfaces (elemental metal systems). This is undoubtedly much more efficient than obtaining the properties of SAACs (alloyed metal systems). In the latter case, due to the interaction between the single guest atom and its images, a large supercell of the whole periodic system containing guest atom and host surface needs to be computed. On the contrary, only smallest unit cell is needed to compute the pristine surface features.

To test the predictive power of obtained models, we employ 10-fold cross validation (CV10). The dataset is first split into ten subsets, and the descriptor identification along with the model training is performed using nine subsets. Then the error in predicting properties of the systems in the remaining subset is evaluated with the obtained model[51–53]. The CV10 error is defined as the average value of the test errors obtained for each of the ten subsets. In SISSO over-fitting may occur with increasing dimensionality of the descriptor (i.e., the number of complex

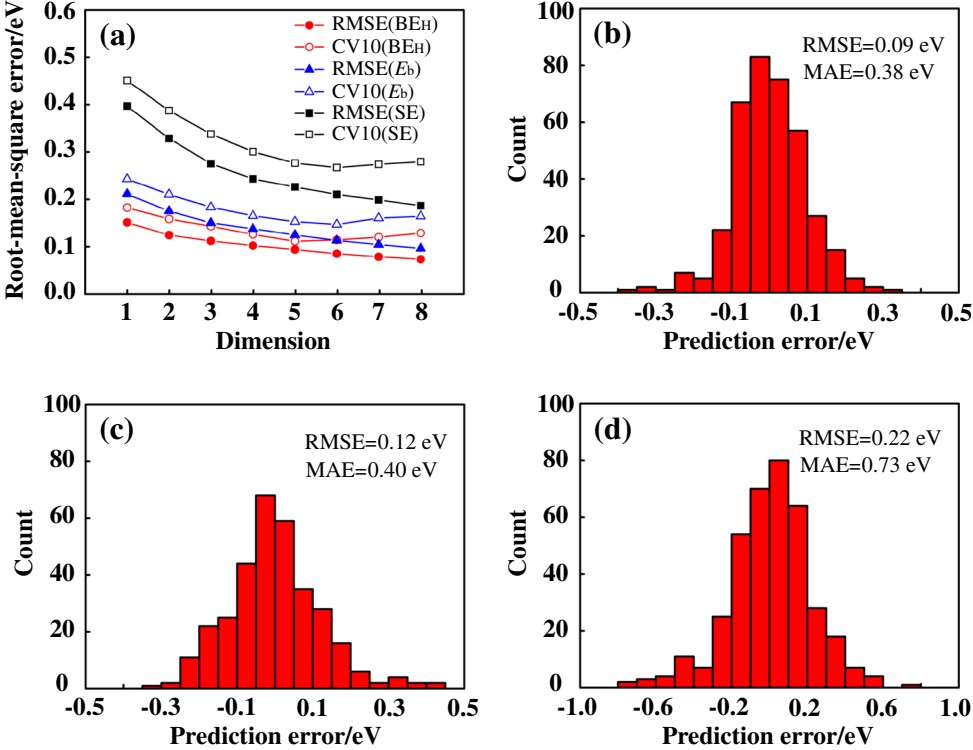

**Fig. 2 SISSO errors and their distribution for different target properties. a** RMSE and the averaged RMSE of the 10 fold cross-validation. **b–d** Distribution of errors for the best models versus RPBE results for $BE_H$ (**b**), $E_b$ (**c**), and SE (**d**).

**Table 2 The identified descriptors and the coefficients and correlations in corresponding SISSO models for $BE_H$, $E_b$, and SE.**

| Property | $d^m$ | Descriptor | Coefficient | Correlation |
|---|---|---|---|---|
| $BE_H$ | $d_1^5$ | $(EA^* + 2F^* - EC) \cdot DT^* \cdot EH^*/(EC^* + F^*)$ | 0.12653E+00 | 0.8964 |
| | $d_2^5$ | $\sqrt[3]{DC} \cdot H^* \cdot DT^* \cdot (|EA^* - EH^*| - |EC - EC^*|)$ | −0.20440E−02 | 0.5891 |
| | $d_3^5$ | $|EH^* - L^* - |EH - F^*|| / (DC^2 + EC \cdot EC^*)$ | −0.50891E+00 | 0.4850 |
| | $d_4^5$ | $|EH - F^* - EH^*| - |EC^* - EC - |DT^* - F^*||$ | 0.34705E−01 | 0.3849 |
| | $d_5^5$ | $L \cdot EC \cdot (EA^* + DS^* - |H - EH| / |L^* - EH^*|$ | −0.48772E−04 | 0.3862 |
| $E_b$ | $d_1^6$ | $((IP^* - L) - |EC^* - DT^*|)/|EC/DC - L^*/IP^*|$ | −0.87339E−01 | 0.7643 |
| | $d_2^6$ | $(EA^* + DC^* + |DC - DT^*|)/(EA^* + EH^* + |L^* - F^*|)$ | −0.19577E−01 | 0.5726 |
| | $d_3^6$ | $(DC + EH^*) \cdot (EC^* - F^*) \cdot (|L - EC| - |EC - EH|)$ | −0.13173E−01 | 0.4568 |
| | $d_4^6$ | $(DT^* - EH) \cdot DC \cdot (H/EC + EA^*/L^*)/EC^*$ | −0.19172E−01 | 0.4414 |
| | $d_5^6$ | $e^{EC} \cdot EH \cdot DS^*/((L^* - DS^*) + |H^* - EC^*|)$ | 0.33549E−01 | 0.3768 |
| | $d_6^6$ | $DC^2 \cdot (EC^* - F^*)/(DT^* - F^* - EA + EC)$ | −0.14362E−02 | 0.3643 |
| SE | $d_1^6$ | $(EC + IP + |F^* - DT^*|) / (IP^*/R + H^*/dd^*)$ | −0.82665E+00 | 0.8969 |
| | $d_2^6$ | $|DC - EB^*| \cdot (L - DC - EC)/EB^2$ | 0.30742E+00 | 0.5346 |
| | $d_3^6$ | $||EC^* - L^*| + |DC - DS^*| - |DC - F^*| - |EC - F^*||$ | 0.11317E+00 | 0.5386 |
| | $d_4^6$ | $|H - IP - L + IP^*| / ((DC/EC) + (EC/H))$ | 0.17455E+00 | 0.3913 |
| | $d_5^6$ | $(F^* - EC) \cdot (L^* - DT^* - IP)/(F^* - EB^*)$ | −0.51761E−02 | 0.3982 |
| | $d_6^6$ | $EC^* \cdot DC \cdot (EB^* - L) \cdot (L + L^* - EC - DS^*)$ | −0.80032E−03 | 0.3379 |

features that are used in construction of the linear model)[49]. The descriptor dimension at which the CV10 error starts increasing identifies the optimal dimensionality of the descriptor (details of the validation approach can be found in Supplementary Methods). For the optimal dimensionality, the same set of primary features is found during CV10 in 9, 8, and 8 cases for the SISSO models of $BE_H$, $E_b$, and SE, respectively. The root-mean-square errors (RMSE), together with the CV10 errors of the SISSO models for $BE_H$, $E_b$, and SE are displayed in Fig. 2a. The obtained optimal descriptor dimensionalities for $BE_H$, $E_b$, and SE of the SAACs are 5, 6, and 6, respectively. Distribution of errors for the best models versus RPBE results is displayed in Fig. 2b–d. The RMSE and maximum absolute error (MAE) of the models are

also shown. The error distributions for all the lower-dimensional models relative to the best ones are displayed in Supplementary Figs. 4–6.

From the Table 2 one can see that the $d$-band center features DC, DC*, DT, DT*, DS, and DS* appear in every dimension of the descriptors for $BE_H$ and $E_b$, consistent with the well-established importance of $d$-band center for adsorption at transition-metal surfaces[21,23,36,44]. The cohesive energies of guest (EC) and host (EC*) bulk metals are selected in each dimension of the descriptor for SE. This is due to the fact that the segregation is driven by the imbalance of binding energy between host and guest–host atoms. Interestingly, most of the descriptor components include only simple mathematical operators (+, −, ·, /, ||),

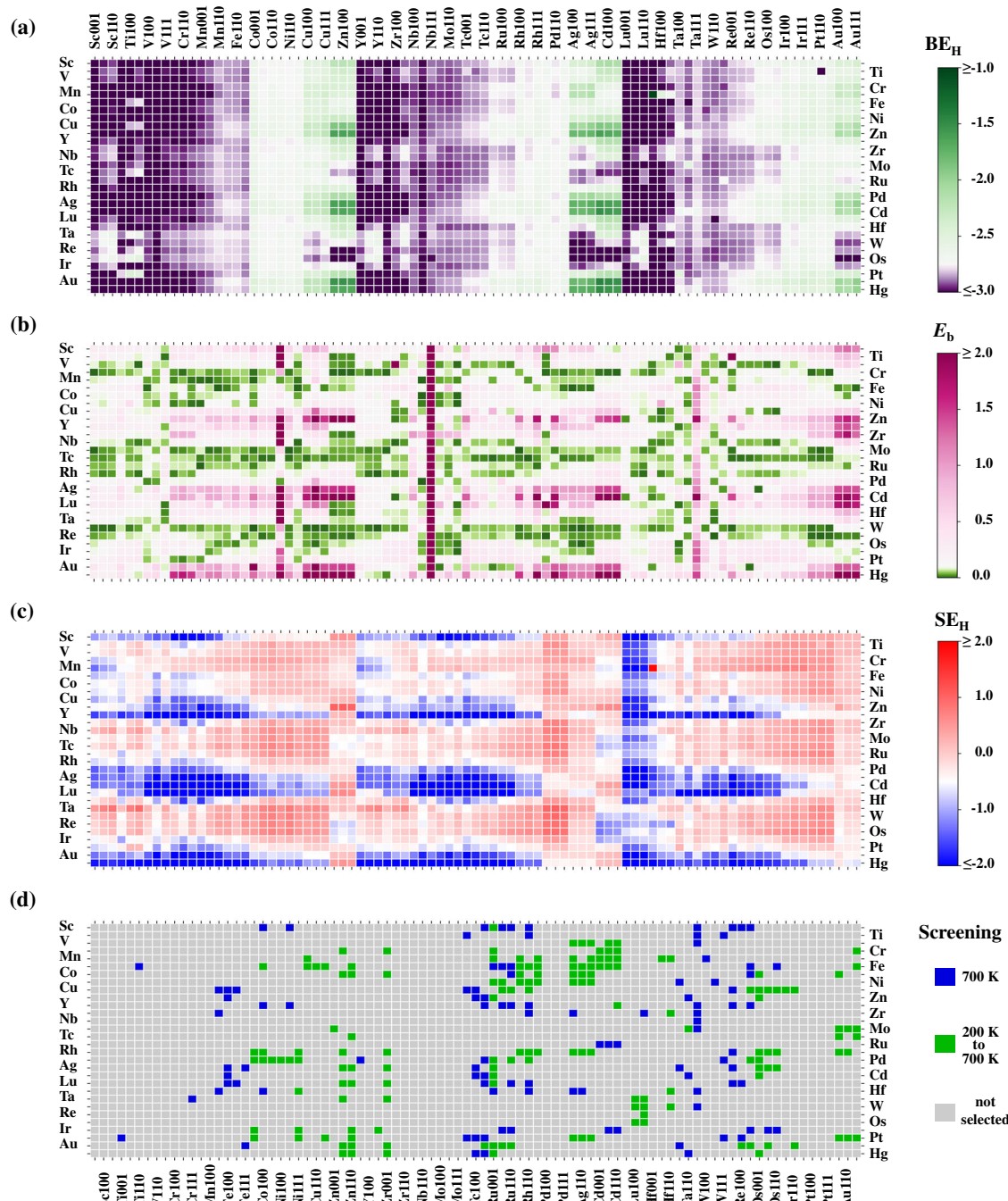

**Fig. 3 Results of high-throughput screening of SAACs with SISSO.** Results for **a** $BE_H$, **b** $E_b$, and **c** $SE_H$ are shown. The promising candidates at different temperatures $T$ are highlighted in **d**. Vertical axis displays the guest atom type, and the horizontal axis displays the host metal surfaces with different surface cuts.

indicating that the primary features already capture most of the complexity of the target properties.

We employ the identified computationally cheap SISSO models to perform high-throughput screening of SAACs to find the best candidates for the hydrogenation reactions. The results for $BE_H$, $E_b$, and $SE_H$ (the segregation energy when surface H adatom is present, where the H adatom induced segregation energy change is included, see the "Methods" part for details) of the flat surfaces are displayed in Fig. 3a–c (see Supplementary Fig. 7 for the results for the stepped surfaces, the values of $BE_H$, $E_b$, and $SE_H$ for all the SAACs are given in Supplementary Data 1).

The choice of the screening criteria for the three properties $BE_H$, $E_b$, and $SE_H$, which are related to the activity and stability of

SAACs, plays the central role in the screening processes and determines the candidates to be chosen. Previous work demonstrates that for the high performance in hydrogenation reactions, SAACs should exhibit weaker binding of H and lower $H_2$ dissociation energy barrier simultaneously[2]. However, different criteria are applicable for different reaction conditions. For example, at low temperatures SAACs can maintain their stability for a longer time. At higher temperatures H atoms will desorb from the surfaces and larger energy barriers can be overcome, resulting in a requirement for stronger binding and higher upper limit of the dissociation barrier $E_b$. Keeping this variability in mind, we consider temperature-dependent and pressure-dependent selection criteria (see "Methods" section below for details on the

selection criteria). We have screened more than five thousand SAAC candidates (including about the same number of flat and stepped surfaces; the values of the primary features for all the candidates can be found in the Supplementary Data 2) at both low temperature (200 K) and high temperature (700 K) at partial $H_2$ pressure $p = 1$ atm. We find 160 flat-surface SAACs (Fig. 3d, in green) and 134 stepped-surface SAACs (Supplementary Fig. 7d, in green) that are both active and stable at a low temperature (200 K). At a higher temperature (700 K), 102 flat-surface SAACs (Fig. 3d, in blue and green) and 136 stepped-surface SAACs (Supplementary Fig. 7d, in blue and green) are classified as promising SAACs for hydrogenation reactions. Moreover, we have identified the SAACs that are promising in a wide range of temperatures (green squares in Fig. 3d for flat surfaces and Supplementary Fig. 7d for stepped surfaces).

Note that, without the stability selection criterion based on $SE_H$, all experimentally established SAACs (Pd/Cu, Pt/Cu, Pd/Ag, Pd/Au, Pt/Au, Pt/Ni, Au/Ru, and Ni/Zn) are predicted to be good catalysts in the temperature range of 200 K < T < 700 K, which is further confirmed by DFT calculations. However, some of these systems (Pd/Ag and Pd/Au) are experimentally shown to have low stability[12,16]. Thus, inclusion of the stability-related property $SE_H$ is of immense importance for a reliable prediction of catalytic performance, as is confirmed by our results. We note that a machine-learning study on stability of single-atom metal alloys has recently been reported[54]. However, our analysis takes into account effects of adsorbates on the segregation energy, which has not been considered previously. For example, the SE for Pd/Ag (110) and Pt/Ag(110) systems are 0.33 eV and 0.46 eV, respectively, implying that the Pd and Pt impurities tend to segregate into the bulk of the Ag(110) systems. However, $SE_H$ for Pd/Ag (110) and Pt/Ag(110) systems are −0.10 eV and −0.21 eV, respectively, suggesting Pd and Pt impurities will segregate to the surface in the presence of H adatom. These results are also consistent with the experimental observations that the efficiency of Pd/Ag single-atom catalysts towards the selective hydrogenation of acetylene to ethylene was highly improved with the pretreatment of the samples under $H_2$ conditions[16].

We define an activity (or efficiency) indicator involving both the free energy of H adsorption ($\Delta G$) and the energy barrier ($E_b$) as $\sqrt{\Delta G^2 + E_b^2}$ to construct an activity-stability map. As shown in Fig. 4, some of the new discovered candidates (bottom-left corner of activity-stability map) are predicted to have both higher stability and efficiency than the reported ones, making them optimized for practical applications (see Supplementary Fig. 8 for the results for the stepped surfaces). As expected, stability and activity are inversely related, which can be seen from the negative slope of the general trend in Supplementary Fig. 8 (showing selected materials) and Supplementary Fig. 9 (showing all explored materials), as well as a cut-off in population of the lower left-hand corner of these plots. Nevertheless, we have found several materials that are predicted to be better SAACs than the so-far reported ones. Considering stability, activity, abundance, and health/safety, two discovered best candidates Mn/Ag(111) and Pt/Zn(0001) are highlighted in Fig. 4. The aggregation energies for Mn/Ag(111), Pt/Zn(0001), and the experimentally established SAACs are also tested and displayed in Supplementary Table 9.

Although the SISSO models are analytic formulas, the corresponding descriptors are complex, reflecting the complexity of the relationship between the primary features and the target properties. While potentially interpretable, the models do not provide a straightforward way of evaluating relative importance of different features in actuating desirable changes in target properties. To facilitate physical understanding of the actuating mechanisms, we apply the subgroup discovery (SGD) approach[55–60]. SGD finds local

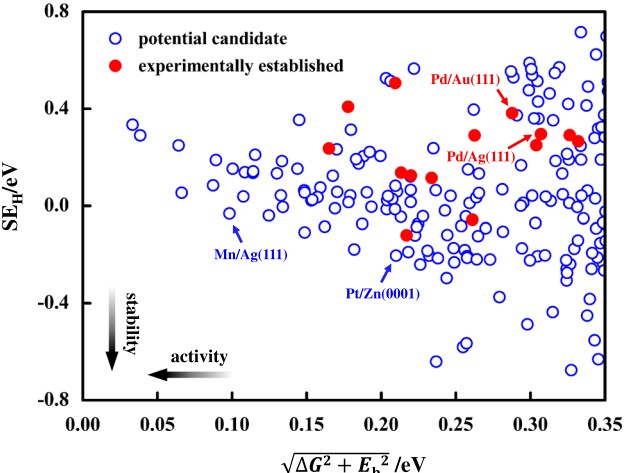

**Fig. 4 Stability vs. activity map for flat SAACs surfaces at $T = 298$ K and $p = 1$ atm.** The $SE_H$ on $y$-axis represents stability and activity parameter $\sqrt{\Delta G^2 + E_b^2}$ is shown on $x$-axis. Experimentally established SAACs are denoted with red solid spheres and the blue open circles represent new predicted candidates.

patterns in the data that maximize a quality function. The patterns are described as an intersection (a selector) of simple inequalities involving provided features, e.g., (feature1 < a1) AND (feature2 > a2) AND…. The quality function is typically chosen such that it is maximized by subgroups balancing the number of data points in the subgroup, deviation of the median of the target property for the subgroup from the median for the whole data set, and the width of the target property distribution within the subgroup[60].

Here, we apply SGD in a novel context, namely as an analysis tool for symbolic regression models, including SISSO. The primary features that enter the complex SISSO descriptors of a given target property are used as features for SGD (see Table 2). The data set includes all 5200 materials and surfaces used in the high-throughput screening. The target properties are evaluated with the obtained SISSO models. Five target properties are considered: $\sqrt{\Delta G^2 + E_b^2}$, SE, $SE_H$, $E_b$, $|\Delta G|$, and $BE_H$. Since we are interested mainly in catalysts that are active at normal conditions, $\Delta G$ is calculated at $T = 300$ K. Our goal is to find selectors that minimize these properties within the subgroup. Such selectors describe actuating mechanisms for minimization of a given target property. For SE, the following best selector is found: ($EC^* \leq −3.85$ eV) AND ($−3.36$ eV $< EC \leq −0.01$ eV) AND (IP $\geq 7.45$ eV). The corresponding subgroup contains 738 samples (14% of the whole population), and the distribution of SE within the subgroup is shown in Supplementary Fig. 10. Qualitatively, the first two conditions imply that the cohesive energy of the host material is larger in absolute value than the cohesive energy of the guest material. Physically this means that bonding between host atoms is preferred over bonding between guest atoms and therefore over intermediate host–guest binding. This leads to the tendency of maximizing the number of host–host bonds by pushing guest atom to the surface. We note that this stabilization mechanism has been already discussed in literature[61], and here we confirm it by data analysis. In addition, we find that stability of SAACs requires that the ionization potential of the guest atom is high. This can be explained by the fact that lower IP results in a more pronounced delocalization of the $s$ valence electrons of the guest atom, and partial charge transfer to the surrounding host atoms. The charge transfer favors larger number of neighbors due to increased Madelung potential, and therefore destabilizes surface position of the guest atom.

We calculate $SE_H$ using SISSO models for SE and $BE_H$ [see Eq. (3) in the "Methods" section]. Therefore, SGD for $SE_H$ is performed using primary features present in the descriptors of both SE and $BE_H$. The top subgroup contains features related to binding of H to the host and guest metal atoms, e.g., $(EB^* < -5.75\,\mathrm{eV})$ AND $(EH^* \leq -2.10\,\mathrm{eV})$ AND $(EH \geq -2.88\,\mathrm{eV})$ AND $(IP^* \leq 7.94\,\mathrm{eV})$ AND $(IP > 8.52\,\mathrm{eV})$ AND $(R \geq 1.29\,\text{Å})$. However, the distribution of SE for this subgroup is very similar to the distribution of $SE_H$, which means that the stability of guest atoms at the surface is weakly affected by H adsorption when guest atoms are already very stable at the surface. The important effect of H adsorption is revealed when we find subgroups minimizing directly $SE_H$—SE (in this case only primary features that appear in the SISSO descriptor of $BE_H$ are considered for SGD analysis). The top subgroup we found contains 72 samples (1.4% of the whole population) and is described by several degenerate selectors, in particular $(-2.35\,\mathrm{eV} \leq EH^* \leq -2.32\,\mathrm{eV})$ AND $(EC^* > -2.73\,\mathrm{eV})$ AND $(EC < -5.98\,\mathrm{eV})$ AND $(H \geq -5.12\,\mathrm{eV})$. This is a very interesting and intuitive result. Distributions of $SE_H$ and SE for this subgroup are shown in Supplementary Fig. 11. The SE for all materials in the subgroup is above 0 eV. However, $SE_H$ is much closer to 0 eV, and is below 0 eV for a significant number of materials in this subgroup. The conditions on the cohesive energy of guest and host metals (very stable bulk guest metal and less stable bulk host metal) are reversed with respect to SE, i.e., adsorption of hydrogen affects strongly the systems where guest atom is unstable at the surface. This increases the reactivity of the guest atom towards an H atom. The condition $(EH^* \geq -2.35\,\mathrm{eV})$ selects materials where interaction of H with a host atom is not too strong, so that H can bind with the guest atom and stabilize it at the surface. The condition $(EH^* \leq -2.32\,\mathrm{eV})$ makes the subgroup narrower, which further decreases median difference $SE_H$—SE but has no additional physical meaning. The condition $(H \geq -5.12\,\mathrm{eV})$ has a minor effect on the subgroup.

One of the top selectors (among several describing very similar data subsets) for minimizing $\sqrt{\Delta G^2 + E_b^2}$ (calculated at $T = 300$ K) is: $(-2.85\,\mathrm{eV} \leq DC \leq 1.95\,\mathrm{eV})$ AND $(DT^* \leq -0.17\,\mathrm{eV})$. The corresponding subgroup contains 1974 samples (38% of the whole population). The distribution of $E_b$ within the subgroup is shown in Supplementary Fig. 10. The selector implies that systems providing low barrier for $H_2$ dissociation, and at the same time balanced binding of H atoms to the surface are characterized by (i) $d$-band center of the bulk guest metal around the Fermi level and (ii) $d$-band center of the host surface top layer below the Fermi level. This can be understood as follows. Condition (i) implies that there is a significant $d$-electron density that can be donated to the adsorbed $H_2$ molecule, facilitating its dissociation. A very similar (apart from slightly different numerical values) condition appears in the selector for the best subgroup for $E_b$ target property alone $[(-2.05\,\mathrm{eV} \leq DC \leq 1.46\,\mathrm{eV})$ AND $(EC^* \geq -6.33\,\mathrm{eV})]$. Condition (ii) implies that the surface $d$-band is more than half-filled, so that additional electrons are available for transferring to the $H_2$ molecule for its activation without causing excessive binding and therefore minimizing $|\Delta G|$ in accordance with Sabatier principle. Indeed, several subgroups of surfaces binding H atoms strongly (minimizing $BE_H$) are described by selectors including condition $DT^* > -0.17$, which is exactly opposite to condition (ii). Analysis of $BE_H$ and $|\Delta G|$ also shows that the strong and intermediate binding of H atoms to the surface is fully controlled by the features of host material.

We note that SGD is capable of finding several alternative subgroups, corresponding to different mechanisms of actuating interesting changes in target properties. These subgroups have a lower quality according to the chosen quality function, but they still contain useful information about a particular mechanism. In fact, they can be rigorously defined as top subgroups under additional constraint of zero overlap (in terms of data points) with previously found top subgroups. Analysis of such subgroups can be a subject of future work. We also note that quality function used in SGD is a parameter and can affect the found subgroups. It should be chosen based on the physical context of the problem. Exploring the role of different factors in the quality function and taking into account proposition degeneracy (no or minor effect of different conditions in the selectors due to correlation between the features) can significantly improve interpretability of the selectors. The interpretability also depends crucially on our physical understanding of the features and relations between them. Nevertheless, in combination with human knowledge SGD analysis allows for development of understanding, that would not be possible without the help of artificial intelligence.

In summary, by combining first-principles calculations and the data-analytics approach SISSO, we have identified accurate and reliable models for the description of the hydrogen binding energy, dissociation energy, and guest-atom segregation energy for SAACs, which allow us to make fast yet reliable prediction of the catalytic performance of thousands SAACs in hydrogenation reactions. The model correctly evaluates performance of experimentally tested SAACs. By scanning more than five thousand SAACs with our model, we have identified over two hundred new SAACs with both improved stability and performance compared to the existing ones. We have also introduced a novel approach to a qualitative analysis of complex SISSO descriptors using data-mining method subgroup discovery. It allows us to identify actuating mechanisms for desirable changes in the target properties, e.g., reaction barrier reduction or an increase in catalyst's stability, in terms of basic features of the material. Our methodology can be easily adapted to designing new functional materials for various applications.

## Methods

All first-principles calculations are performed with the revised Perdew-Burke-Ernzerhof (RPBE) functional[62] as implemented in the all-electron full-potential electronic-structure code FHI-aims[63]. The choice of functional is validated based on a comparison of calculated $H_2$ adsorption energies to the available experimental results[64] (see Supplementary Table 1). Nevertheless, it is expected that, because of the large set of systems inspected and the small variations introduced by the functional choice, the main trends will hold even when using another functional (see Supporting Information for more details on the computational setup). The climbing-image nudged elastic band (CI-NEB) algorithm is employed to identify the transition state structures[65].

$BE_H$ are calculated using Eq. (1), where $E_{H/support}$ is the energy of the total H/support system, $E_{support}$ is the energy of the metal alloy support, and $E_H$ is the energy of an isolated H atom.

$$BE_H = E_{H/support} - E_{support} - E_H \qquad (1)$$

The surface segregation energy in the dilute limit, SE, is defined as the energy difference of moving the single impurity from the bulk to the surface. In this work, it is calculated using Eq. (2), where $E_{top\text{-}layer}$ and $E_{nth\text{-}layer}$ correspond to the total RPBE energies of the slab with the impurity in the top and $n$th surface layer, respectively. The value of $n$ is chosen so that the energy difference between $E_{nth\text{-}layer}$ and $E_{(n-1)th\text{-}layer}$ is less than 0.05 eV.

$$SE = E_{surface} - E_{nth-layer} \qquad (2)$$

The surface segregation energy when surface H adatom is present (the H is put at the most stable adsorption site for each system), $SE_H$, is calculated using Eq. (3).

$$SE_H = SE + \Delta E_H, \qquad (3)$$

where $\Delta E_H = BE_{H\text{-top-layer}} - BE_{H\text{-pure}}$ is the H adatom-induced segregation energy change.

Here $BE_{H\text{-top-layer}}$ and $BE_{H\text{-pure}}$ are the hydrogen adatom binding energies with the impurity in the top layer and the $BE_H$ of the pure system without impurity. Thus, the $SE_H$ can be derived from the models of SE and $BE_H$.

Using first-principles inputs as training data, we have employed SISSO to single out a physically interpretable descriptor from a huge number of potential candidates. In practice, a huge pool of more than 10 billion candidate descriptors is first constructed iteratively by combining user-defined primary features with a set of

mathematical operators. The number of times the operators are applied determines the complexity of the resulting descriptors. We consider up to three levels of complexity (feature spaces) $\Phi_1$, $\Phi_2$, and $\Phi_3$. Note that a given feature space $\Phi_n$ also contains all of the lower rung (i.e., $n − 1$) feature spaces. Subsequently, the desired low-dimensional representation is obtained from this pool[49]. The details of the feature space ($\Phi_n$) construction and the descriptor identification processes can be found in the Supplementary Methods. The proper selection of primary features is crucial for the performance of SISSO-identified descriptors. Inspired by previous studies[31,38], we consider three classes of primary features (see Table 1) related to the metal atom, bulk, and surface. The more detailed description and values of all the primary features are given in the Supplementary Table 2, Supplementary Table 3, Supplementary Data 1, and Supplementary Data 2.

The selection of the promising candidates at various temperatures and hydrogen partial pressures is performed based on ab initio atomistic thermodynamics[66]. H adsorption/desorption on SAAC surfaces as a function of temperature and $H_2$ partial pressure $(T, p)$ is characterized by the free energy of adsorption $\Delta G$:

$$\Delta G = E_{H/support} − E_{support} − \mu_H(T, p) \tag{4}$$

with the chemical potential of hydrogen $\mu_H = \frac{1}{2}\mu_{H_2}$ obtained from:

$$\mu_H = \frac{1}{2}\left(E_{H_2} + \Delta\mu_{H_2}(T, p)\right), \tag{5}$$

where $\Delta\mu_{H_2}(T, p) = \mu_{H_2}(T, p^0) − \mu_{H_2}(T^0, p^0) + k_B T \ln(\frac{p}{p^0})$.

Here $T^0 = 298$ K and $p^0 = 1$ atm. The frst two terms are taken from JANAF thermochemical tables[67]. In the following, we set $p = 1$ atm.

According to Sabatier principle the optimum heterogeneous catalyst should bind the reactants strong enough to allow for adsorption, but also weak enough to allow for the consecutive desorption[25]. In this work, a $BE_H$ range is defined by the conditions:

$$|BE_H − \frac{1}{2}(E_{H_2} − 2E_H) − \frac{1}{2}\Delta\mu_{H_2}(T)| < 0.3 \text{ eV}, \tag{6}$$

where $E_{H_2} − 2E_H$ is the hydrogen binding energy of the hydrogen molecule. The experimental value of $−4.52$ eV[68] was used in this work.

The above conditions correspond to the free-energy bounds:

$$|\Delta G| < 0.3 \text{ eV}, \tag{7}$$

Conditions on energy barrier ($E_b$) are defined by considering Arrhenius-type behaviour of the reaction rate on $E_b$ and $T$. Assuming that acceptable barriers are below 0.3 eV for $T^0 = 298$ K, we estimate acceptable barrier at any temperature as:

$$E_b < \frac{0.3 T}{T^0} \text{ eV}. \tag{8}$$

Similarly the bounds for $SE_H$ are determined by imposing a minimum 10% ratio for top-layer to subsurface-layers dopant concentration by assuming an Arrhenius-type relation with $SE_H$ interpreted as activation energy:

$$SE_H < k_B T \ln(10). \tag{9}$$

The subgroup discovery was performed using RealKD package (https://bitbucket.org/realKD/realkd/). Each feature was split to 15 subsets using 15-means clustering algorithm. The borders between adjacent data clusters (a1, a2,…) are applied further for construction of inequalities (feature1 < a1), (feature2 ≥ a2), etc. While final result might depend on the number of considered clusters, in our previous study we found that relatively high numbers of considered clusters provide essentially the same result[60]. The candidate subgroups are built as conjunctions of obtained simple inequalities. The main idea of SGD is that the subgroups are unique if the distribution of the data in them is as different as possible from the data distribution in the whole sampling. Here the data distribution is the distribution of a target property ($\sqrt{\Delta G^2 + E_b^2}$, SE, $E_b$, $|\Delta G|$, and $BE_H$). The uniqueness is evaluated with a quality function. In this study we used the following function:

$$Q(S) = \frac{s(S)}{s(P)}\left(\frac{med(P) − med(S)}{med(P) − min(P)}\right)\left(1 − \frac{amd(S)}{amd(P)}\right) \tag{10}$$

with S—subgroup, P—whole sampling, s—size, med and min—median and minimal values of a target property, amd—absolute average deviation of the data around the median of target property. With this function the algorithm is searching for subgroups with lower values of target properties. The search was done with an adapted for such purposes Monte-Carlo algorithm[59], in which first a certain number of trial conjunctions (seeds) is generated. Afterwards, for each seed (accompanied with pruning of inequalities) the quality function is calculated. We have tested here several numbers of initial seeds: 10,000, 30,000, 50,000, and 100,000. The subgroups with the overall high quality function value were selected.

## Data availability
All relevant data are available from the corresponding authors upon reasonable request.

## Code availability
FHI-aims: https://aimsclub.fhi-berlin.mpg.de.
SISSO: https://github.com/rouyang2017/SISSO.
SGD: https://bitbucket.org/realKD/realkd/.

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

## Acknowledgements

S.V.L. is supported by Skolkovo Foundation Grant. The machine-learning methodology development was funded by RFBR and INSF, project number 20-53-56065. Y.G. is supported by the National Natural Science Foundation of China (11604357, 11574340). R.O. is supported by the National Key Research and Development Program of China (2018YFB0704400) and the Program of Shanghai Youth Oriental Scholars.

## Author contributions

S.V.L. created the idea and conceived the work. S.V.L. and Y.G. designed and supervised the project. S.V.L. and A.M. supervised the SGD analysis. R.O. supervised the SISSO analysis. Z.-K.H. performed all the calculations. Z.-K.H. and D.S. wrote the manuscript with inputs from all the authors. All authors contributed to the analysis and interpretation of the results. All the authors commented on the manuscript and have given approval to the final version of the manuscript.

## Competing interests

The authors declare no competing interests.
