## [Peer Review File · Nature Communications]

Reviewers' comments:

Reviewer #1 (Remarks to the Author):

This interesting work leverages the recently developed SISO (sure independence screening and sparsifying operator) algorithm to develop descriptors of stability and activity for screening single atom alloy catalysts (SAACs). The main impact of the work seems to be in identifying a number of new SAACs for potential experimental study. Two SAACs (Mn/Ag(111) and Pt/Zn(0001)) are identified as particularly promising. The paper is application-based in nature and doesn't appear to contribute significant conceptual advances for SAACs or machine learning applications. While the work seems to be well done and the paper is well-written, and the SAAC and ML topics are of great interest these days, the paper scope is not particularly ambitious. This work may be suitable for Nat. Comm. if its scope is slightly broadened and fundamental insight is improved. My comments and suggestions are provided below.

- Page 3: Figure 1a shows the H-atom binding energy vs. the d-band center for the Ag(110) host surface. The d-band center is typically calculated from the projected DOS, and it is unclear which surface atom the d-states are being taken from based on the text. Please clarify. Furthermore, using the Ag(110) surface to show that the d-band model is broken because of SAACs is not very convincing. Ag(110) should not follow the d-band model due to the fact that its d-band is completely full; therefore binding trends on Ag(110) should depend more on changes in sp electron density and Pauli repulsion (see DOI: https://link.springer.com/chapter/10.1007/978-94-015-8911-6_11 for details on why Ag(110) should not follow the d-band model). Do you have other examples of SAAC breaking scaling relations besides a host metal with a full d-band?

- Page 6: The manuscript may benefit from some discussion on the robustness of the SISO models identified in Table 2. Does adding or removing a training data point lead to a different descriptor? If so, does the new descriptor still yield similar model behavior?

- Page 6: Can you comment at all on the relative importance of the primary features included in the model relative to their impact on the overall prediction? This may lead to interesting general conclusions, like elucidating whether the primary features of the guest or host metal play a larger role for a given target property. Discussing the relative importance of guest vs. host metal features would be quite informative.

- Page 7: It seems the authors identify Tc alloys as promising SAACs. It's worth noting that there may be other health/safety considerations when using Tc in catalytic applications due to the fact that all Tc isotopes are radioactive.

- Page 8: I believe that the manuscript would benefit from an expanded discussion of Figure 3 that explains the general trends that emerge from the high-throughput screening results (e.g., in general, what types of guest atoms yield SAACs with low hydrogen dissociation barriers? What guest/host combinations lead to small segregation energies and why in terms of atomic radii size or other features?)

Minor Comments Main Text

- Figure 1: If you change solid red circles to be different symbols for hollow bridge, bridge, top that would be more information-rich and potentially informative (just a suggestion).
- Table S1 caption. "the surface-based primary features were calculated using the slab unit cell consisting of one atom per atomic layer." Should be "The".
- Page 6: The text indicates that the primary features DC, DC*, DT, DT*, DS, and DS* appear in every dimension of the descriptors for hydrogen binding energy and dissociation barrier. However, based on Table 1, it is unclear what the DT and DS primary features are as opposed to the DT* and DS* primary features. From reading the SI, it seems * denotes host metal from guest atom feature. I think this * notation can be clarified in Table 1.
- Page 9: "higher stability and efficiency than the reported ones, making them perfectly optimized for practical applications." Perfectly optimized seems to be a strong choice of words here. Perhaps remove the word "perfectly".

Minor Comments on Supporting Information

- Page 1: "Spin-polarization effects are tested for and included where appropriate." Is it noted somewhere for which spin polarization effects are included? This is a vague statement and could perhaps be made more explicit
- Figure S1 caption. "bcc(110) e," should be bcc(110) (e)
- Table S3: "Binding energy of host metal dimers", So this is a dimer energy for $A(g) + A(g) \rightarrow A_2(g)$? Could perhaps be clarified.
- Font size for the captions in Figures S3-S5 are smaller than the other Figure S captions (i.e., font size 10 vs. 12).
- Table S5: "Number of system with the predicted and calculated segregation energy meet the same condition of $SE < kT \ln(10) (N_{meet}) \dots$ " Perhaps it should read as "Number of systems with the predicted and calculated segregation energies that meet the same condition..."

Reviewer #2 (Remarks to the Author):

The manuscript presents machine learning models of single atom catalysts and screening procedure for design of hydrogenation catalysts based on this new type of alloys emerged in recent years. The features designed are easily available properties that are tabulated including electronic structure, bulk properties, etc. The target properties include the binding energy, activation barrier and the segregation. Those properties are crucial for screening high performance hydrogenation catalysts. While the work is thoroughly done in those aspects, this does not reach the standard of Nat Commu.

1. The novelty of the approach is lacking. Compressed sensing is used recently in M. Andersen, S. V. Levchenko, M. Scheffler, K. Reuter, Beyond Scaling Relations for the Description of Catalytic Materials. ACS Catal. 9, 2752–2759 (2019).
2. While the SISSO with cross validation is reasonably accurate for training a small dataset, its generalization to new systems is still the biggest problems for all current learning framework. Active learning approach was used to tackle this problem (K. Tran, Z. W. Ulissi, Active learning across intermetallics to guide discovery of electrocatalysts for CO₂ reduction and H₂ evolution. Nature Catalysis. 1, 696–703 (2018).), while a large amount of calculations are required. The current study used only ~300 datapoints for training and extend the model to ~5000 space without validation of model prediction.
3. The criteria for screening catalysts used in this study is arbitrary. Without detailed kinetics, the approach can only provide a rough screening of candidate materials.
4. For segregation, a recent study by Grabow et al. (K. K. Rao, Q. K. Do, K. Pham, D. Maiti, L. C. Grabow, Extendable Machine Learning Model for the Stability of Single Atom Alloys. Top. Catal. (2020), doi:10.1007/s11244-020-01267-2.). Even the *H binds weakly on the metals, its effect on the segregation is not considered in any of those studies.
5. The most fundamental problem of this study and the approach in general is their lacking of understanding the uniqueness of single atom alloys. Although the SISSO method comes up formula in reduced feature space, the physics is missing. The message to the community by the study is rather incremental while does not provide a way forward to tackle all those issues.

Reviewer #3 (Remarks to the Author):

The authors report the use of modern data analytics towards the reliable prediction of activity and stability of dilute alloy "single atom catalysts" for hydrogenation. The topic of particular interest as single atom catalysts have made massive strides for oxidation reactions but have had limited success for reductions particularly due to lack of activity and/or abysmal stability. The strength of the authors approach is that it addresses catalyst screening beyond the simple approximation BEP, d-band center etc. etc. etc. These concepts are embedded in the psyche of computational catalysis so deep that we forget they are simple models and, in many instances, too simple for quantitative predictions-but excellent for rationalizations on small data sets.

The authors show that by assembling a large number of atomic, bulk and alloy descriptors (table 1) they are able to perform a high dimensional correlation with the ab initio data to yield property predictions FAR more accurate than the existing simple concepts. On the one hand this is a great step forward for screening studies on the other hand if I have a more complex fitting function, I do expect a better fit. The one worry I have is this then becomes a brute force approach without the intellectual understanding that can be provided by a simple model. In this respect it might have been more intellectually pleasing for the authors to consider if there was a smaller subset of parameters (2-3) that might do a reasonable job (better than linear fits but not the full-blown set) which might hint at a simpler model. As is, the approach is fine I do worry about both overfitting/underfitting of data but do believe the authors have covered this ground adequately.

Finally, then the result of this study is that using their model they can rapidly predict the results of DFT calculations and use that data to make predictions about activity and stability based on simple energetic parameters such as presented in Figure 4. In my opinion this is the most important plot in the whole paper and the authors did not really deal with its ramifications very well. The wisdom in single atom catalysts (particularly for hydrogenation) is that the more active the species the less stable it will be - hence the scarcity of single atoms (dilute alloys) that are reported. If the authors are correct there is a large abundance of materials far in the lower right-hand corner (active and stable) that should break this trend whereas those that do exist are mostly in the upper right-hand corner (active but less stable). This is the most significant discovery/prediction in the paper as far as I am concerned, and the authors barely comment on it. Sadly, a follow-on experimental study making targets and validating the prediction would be a breakthrough and this is also not done.

Ultimately my problem is that screening for screening sake, without understanding new things, and without verifying that my parameters to define the screening criteria are valid is a reasonable technical accomplishment and not consistent with an advance I would expect in a Nature Journal. I think this paper would be highly appropriate for a journal such as ACS Catal or a chem-informatics journal but other than a more advanced fitting procedure for predicting DFT data I see no real advance here.

Response to Referees

Reviewer 1: This interesting work leverages the recently developed SISO (sure independence screening and sparsifying operator) algorithm to develop descriptors of stability and activity for screening single atom alloy catalysts (SAACs). The main impact of the work seems to be in identifying a number of new SAACs for potential experimental study. Two SAACs (Mn/Ag(111) and Pt/Zn(0001)) are identified as particularly promising. The paper is application-based in nature and doesn't appear to contribute significant conceptual advances for SAACs or machine learning applications. While the work seems to be well done and the paper is well-written, and the SAAC and ML topics are of great interest these days, the paper scope is not particularly ambitious. This work may be suitable for Nat. Comm. if its scope is slightly broadened and fundamental insight is improved. My comments and suggestions are provided below.

Response: We thank the reviewer for the positive comments on our work. Indeed, by identifying the model with both the activity and stability parameters of the SAACs we could confirm the experimentally studied high performance SAACs. Moreover, we predict two new particularly promising systems. Keeping the reviewer's suggestions in mind, in the revised manuscript we now analyze the correlations of each component of the selected best descriptor with the target properties and discuss their physical significance. We also highlight the importance of using the combination of features rather than focusing on individual feature's role in the description of the target properties. Thus, we have stepped beyond the well-established *d*-band center theory, scaling relationships, and the Brønsted-Evans-Polanyi relationship, and have focused on the importance of data analytics in finding new SAACs.

1) Page 3: Figure 1a shows the H-atom binding energy vs. the *d*-band center for the Ag(110) host surface. The *d*-band center is typically calculated from the projected DOS, and it is unclear which surface atom the *d*-states are being taken from based on the text. Please clarify. Furthermore, using the Ag(110) surface to show that the *d*-band model is broken because of SAACs is not very convincing. Ag(110) should not follow the *d*-band model due to the fact that its *d*-band is completely full; therefore binding trends on Ag(110) should depend more on changes in *sp* electron density and Pauli repulsion (see DOI: https://link.springer.com/chapter/10.1007/978-94-015-8911-6_11 for details on why Ag(110) should not follow the *d*-band model). Do you have other examples of SAAC breaking scaling relations besides a host metal with a full *d*-band?

Response: We thank the reviewer for highlighting this important aspect. In the present study, the *d*-band centers are calculated from the *d* orbitals projected on the single guest atom only. Note that, to validate the choice of our *d*-band centers, we have calculated *d*-band centers for the *d* orbitals projected on (i) the single guest atom and its 1st nearest neighbor shell and (ii) the whole slab. However, the correlation between the binding energy and these later two *d*-band centers are found to be worse compared to the *d*-band center of the single guest atom. This is now clarified in the revised manuscript [page 3].

In the revised manuscript we have included the correlations between the binding energy and the *d*-band center for another system as well [Pt(111) surface], whose *d*-band is not completely full.

Changes made:

1. We have replaced the sentence “we first investigate correlation between BE_H and the d -band center for the alloyed systems” by “we first investigate correlation between BE_H and the d -band center for the alloyed systems. Note that, d -band centers are calculated from the d orbitals projected on the single guest atom only. We find that this way of calculating d -band center provides better correlation with other properties than d -band centers for the d orbitals projected on (i) the single guest atom plus it’s 1st nearest neighbor shell or (ii) the whole slab [Topics in Catalysis 61, 462-474 (2018)].” on page 3 of the revised manuscript.
2. We have added Figure S2 by including also the system of Pt(111) surfaces which is reproduced as Figure R1 below.

Figure R1. Correlation between (a) H-atom binding energy BE_H and the d-band center and (b) the H_2 dissociation energy barrier E_b and the H_2 dissociation reaction energy for Pt(111) based SAACs.

2) Page 6: The manuscript may benefit from some discussion on the robustness of the SISO models identified in Table 2. Does adding or removing a training data point lead to a different descriptor? If so, does the new descriptor still yield similar model behavior?

Response: The descriptor is robust and remains unchanged upon randomly removing one training data point. We have randomly kept out one data point for each model and have repeated the process 5 times to check the robustness of the descriptor. Moreover, for the optimal dimensionality, the same set of primary features is found during CV10 in 9, 8, and 8 cases for the SISO models of BE_H , E_b , and SE, respectively. Also, new systems that were not included in the training set were used as test set to further confirm the high transferability of our model. Finally, some of the high-throughput screening selected high performance SAACs, including all the experimentally evidenced systems and

our suggested top two best systems, are validated confirmed by density-functional theory calculations.

Changes made:

We have added the sentence “For the optimal dimensionality, the same set of primary features is found during CV10 in 9, 8, and 8 cases for the SISSO models of BE_H , E_b , and SE, respectively” on page 6 of the revised manuscript.

3) Page 6: Can you comment at all on the relative importance of the primary features included in the model relative to their impact on the overall prediction? This may lead to interesting general conclusions, like elucidating whether the primary features of the guest or host metal play a larger role for a given target property. Discussing the relative importance of guest vs. host metal features would be quite informative.

Response: We thank the referee for this suggestion. In this work we highlight the importance of the combinations of the primary features rather than using each feature individually to describe the target properties. However keeping the referee’s advice in mind, we have now introduced a novel general approach to the analysis of complex symbolic-regression models, based on the data mining approach called subgroup discovery. This have allowed us to uncover physical role of particular features, as well as relative role of guest versus host features.

Changes made:

We have added the following paragraphs on page 10 and 11 of the revised manuscript.

“Although the SISSO models are analytic formulas, the corresponding descriptors are complex, reflecting the complexity of the relationship between the primary features and the target properties. While potentially interpretable, the models do not provide a straightforward way of evaluating relative role of different features in actuating desirable changes in target properties. To facilitate physical understanding of the actuating mechanisms, we apply the subgroup discovery (SGD) approach.⁵⁵⁻⁶⁰ SGD finds local patterns in data that maximize a quality function. The patterns are described as an intersection (a selector) of simple inequalities involving provided features, e.g., $(\text{feature1} < a1)$ AND $(\text{feature2} > a2)$ AND... . The quality function is typically chosen such that it is maximized by subgroups balancing the number of data points in the subgroup, deviation of the median of the target property for the subgroup from the median for the whole data set, and the width of the target property distribution within the subgroup.⁶⁰”

“Here, we apply SGD in a novel context, namely as an analysis tool for symbolic regression models, including SISSO. The primary features that enter the complex SISSO descriptors of a given target property are used as features for SGD (see Table 2). The data set includes all 5200 materials and surfaces used in the high-throughput screening. The target properties are calculated using the obtained SISSO models. Five target properties are considered: $\sqrt{\Delta G^2 + E_b^2}$, SE, SE_H , E_b , $|\Delta G|$, and BE_H . Since we are interested mainly in catalysts that are active at normal conditions, ΔG is calculated at $T = 300$ K. Our goal is to find selectors that *minimize* these properties within the subgroup. Such selectors describe actuating mechanisms for minimization of a given target property. For SE, the following best selector is found: $(EC^* \leq -3.85 \text{ eV})$ AND $(-3.36 \text{ eV} < EC \leq -0.01 \text{ eV})$

AND ($IP \geq 7.45$ eV). The corresponding subgroup contains 738 samples (14% of the whole population), and the distribution of SE within the subgroup is shown in Figure S10. Qualitatively, the first two conditions imply that the cohesive energy of the host material is larger in absolute value than the cohesive energy of the guest material. Physically this means that bonding between host atoms is preferred over bonding between guest atoms and therefore over intermediate host-guest binding. This leads to the tendency of maximizing number of host-host bonds by pushing guest atom to the surface. This stabilization mechanism has been discussed in literature,⁶¹ and here we confirm it by *data analysis*. In addition, we find that stability of SAACs requires that ionization potential of the guest atom is high. This can be explained by the fact that lower IP results in more pronounced delocalization of the *s* valence electrons of the guest atom and partial charge transfer to the surrounding host atoms. The charge transfer favors larger number of neighbors due to increased Madelung potential, and therefore destabilizes surface position of the guest atom.

We calculate SE_H using SISSO models for SE and BE_H [see equation (3) in the Methods section]. Therefore, SGD for SE_H is performed using primary features appearing in the descriptors of both SE and BE_H . The top found subgroup contains features related to binding of H to the host and guest metal atoms, e.g. ($EB^* < -5.75$ eV) AND ($EH^* \leq -2.10$ eV) AND ($EH \geq -2.88$ eV) AND ($IP^* \leq 7.94$ eV) AND ($IP > 8.52$ eV) AND ($R \geq 1.29$ Å). However, the distribution of SE for this subgroup is very similar to the distribution of SE_H , which means that the stability of guest atoms at the surface is weakly affected by H adsorption when the surface guest atoms are already very stable. The important effect of H adsorption is revealed when we find subgroups minimizing directly $SE_H - SE$ (in this case only primary features that appear in the SISSO descriptor of BE_H are considered for SGD analysis). The top subgroup we found contains 72 samples (1.4% of the whole population) and is described by several degenerate selectors, in particular (-2.35 eV $\leq EH^* \leq -2.32$ eV) AND ($EC^* > -2.73$ eV) AND ($EC < -5.98$ eV) AND ($H \geq -5.12$ eV). This is a very interesting and intuitive result. Distributions of SE_H and SE for this subgroup are shown in Figure S11. The SE for all materials in the subgroup is above 0 eV. However, SE_H is much closer to 0 eV, and is below 0 eV for a significant number of materials in this subgroup. The conditions on the cohesive energy of guest and host metals (very stable bulk guest metal and less stable bulk host metal) are reversed with respect to SE, i.e., adsorption of hydrogen affects strongly the systems where guest atom is unstable at the surface. This increases the reactivity of the guest atom towards an H atom. The condition ($EH^* \geq -2.35$ eV) selects materials for which interaction of H with a host atom is not too strong, so that H can bond with the guest atom and stabilize it at the surface. The condition ($EH^* \leq -2.32$ eV) makes the subgroup narrower, which further decreases median difference $SE_H - SE$ but has no additional physical meaning. The condition ($H \geq -5.12$ eV) has a minor effect on the subgroup.

One of the top selectors (among several describing very similar data subsets) for minimizing $\sqrt{\Delta G^2 + E_b^2}$ (calculated at $T = 300$ K) is: (-2.85 eV $\leq DC \leq 1.95$ eV) AND ($DT^* \leq -0.17$ eV). The corresponding subgroup contains 1974 samples (38% of the whole population), and the distribution of E_b within the subgroup is shown in Figure S10. The selector implies that systems providing low barrier for H_2 dissociation and at the same time balanced binding of H atoms to the surface are characterized by (i) *d*-band center of the bulk guest metal around the Fermi level and (ii) *d*-band center of the host surface top layer below the Fermi level. This can be understood as follows.

Condition (i) implies that there is a significant *d*-electron density that can be donated to the adsorbed H₂ molecule, facilitating its dissociation. A very similar (apart from slightly different numerical values) condition appears in the selector for the best subgroup for *E_b* target property alone [$-2.05 \text{ eV} \leq DC \leq 1.46 \text{ eV}$) AND ($EC^* \geq -6.33 \text{ eV}$)]. Condition (ii) implies that the surface *d*-band center is more than half filled, which provides additional electrons for transferring to the H₂ molecule, but without excessive binding, to minimize $|\Delta G|$ in accordance with Sabatier principle. Indeed, several subgroups of strongly bound H atoms (minimizing BE_H) are described by selectors including condition $DT^* > -0.17$, which is exactly opposite to condition (ii). Analysis of BE_H and $|\Delta G|$ also shows that the strong and intermediate binding of H atoms to the surface is fully controlled by the features of host material.

We note that SGD is capable of finding several alternative subgroups, corresponding to different mechanisms of actuating interesting changes in target properties. These subgroups have a lower quality according to the chosen quality function, but they still contain useful information about a particular mechanism. In fact, they can be rigorously defined as top subgroups under additional constraint of zero overlap (in terms of data points) with previously found top subgroups. Analysis of such subgroups can be a subject of future work. We also note that quality function used in SGD is a parameter and can affect the found subgroups. It should be chosen based on the physical context of the problem. Exploring the role of different factors in the quality function and taking into account proposition degeneracy (no or minor effect of different conditions in the selectors due to correlation between the features) allows us to develop an understanding that may not be possible without the SGD analysis.”

4) Page 7: It seems the authors identify Tc alloys as promising SAACs. It's worth noting that there may be other health/safety considerations when using Tc in catalytic applications due to the fact that all Tc isotopes are radioactive?

Response: We agree with the referee that health/safety considerations are very important for catalytic applications. This point is now duly mentioned on page 11 of the revised manuscript.

Changes made:

We have changed the sentence “Considering stability, activity, and abundance, two discovered best candidates Mn/Ag(111) and Pt/Zn(0001) are highlighted in Figure 4” to “Considering stability, activity, abundance, and health/safety, two discovered best candidates Mn/Ag(111) and Pt/Zn(0001) are highlighted in Figure 4” on page 11 of the revised manuscript.

5) Page 8: I believe that the manuscript would benefit from an expanded discussion of Figure 3 that explains the general trends that emerge from the high-throughput screening results (e.g., in general, what types of guest atoms yield SAACs with low hydrogen dissociation barriers? What guest/host combinations lead to small segregation energies and why in terms of atomic radii size or other features?)

Response: We thank the referee for these suggestions. In the revised manuscript, we apply the subgroup discovery (SGD) approach to evaluate relative role of different features in actuating

desirable changes in target properties and to facilitate physical understanding of the actuating mechanisms. Please referee to comment 3) for detailed discussion.

6) Minor Comments Main Text

- Figure 1: If you change solid red circles to be different symbols for hollow bridge, bridge, top that would be more information-rich and potentially informative (just a suggestion).
- Table S1 caption. “the surface-based primary features were calculated using the slab unit cell consisting of one atom per atomic layer.” Should be “The”.
- Page 6: The text indicates that the primary features DC, DC*, DT, DT*, DS, and DS* appear in every dimension of the descriptors for hydrogen binding energy and dissociation barrier. However, based on Table 1, it is unclear what the DT and DS primary features are as opposed to the DT* and DS* primary features. From reading the SI, it seems * denotes host metal from guest atom feature. I think this * notation can be clarified in Table 1.
- Page 9: “Higher stability and efficiency than the reported ones, making them perfectly optimized for practical applications.” Perfectly optimized seems to be a strong choice of words here. Perhaps remove the word “perfectly”.

Minor Comments on Supporting Information

- Page 1: “Spin-polarization effects are tested for and included where appropriate.” Is it noted somewhere for which spin polarization effects are included? This is a vague statement and could perhaps be made more explicit
- Figure S1 caption. “bcc(110) e,” should be bcc(110) (e)
- Table S3: “Binding energy of host metal dimers”, So this is a dimer energy for $A(g) + A(g) \rightarrow A_2(g)$? Could perhaps be clarified.
- Font size for the captions in Figures S3-S5 are smaller than the other Figure S captions (i.e., font size 10 vs. 12).
- Table S5: “Number of system with the predicted and calculated segregation energy meet the same condition of $SE < kT \ln(10)$ (Nmeet)...” Perhaps it should read as “Number of systems with the predicted and calculated segregation energies that meet the same condition...”

Response: We thank the referee for pointing these issues/errors. We have modified all these issues/errors accordingly in the revised manuscript and supporting information.

Reviewer 2: The manuscript presents machine learning models of single atom catalysts and screening procedure for design of hydrogenation catalysts based on this new type of alloys emerged in recent years. The features designed are easily available properties that are tabulated including electronic structure, bulk properties, etc. The target properties include the binding energy, activation barrier and the segregation. Those properties are crucial for screening high performance hydrogenation catalysts. While the work is thoroughly done in those aspects, this does not reach the standard of Nat Comm.

Response: We thank the referee for the critical comments. In the revised manuscript we have applied the subgroup discovery (SGD) approach to evaluate relative role of different features in actuating desirable changes in target properties and to facilitate physical understanding of the actuating mechanisms. The combined SISSO and SGD data analytic approaches are novel which provide us

not only predictive models but also new understanding. This allows us to go beyond the well-established *d*-band center theory, scaling relationships, and the Brønsted-Evans-Polanyi relationship.

1) The novelty of the approach is lacking. Compressed sensing is used recently in M. Andersen, S. V. Levchenko, M. Scheffler, K. Reuter, Beyond Scaling Relations for the Description of Catalytic Materials. ACS Catal. 9, 2752–2759 (2019).

Response: We would like to emphasize the advancement and novelty of our work as follows:

- (i) The aim of our work is to predict potential SAACs for hydrogenation reactions, which are not only active but also stable and thus suitable for several practical applications. It is noteworthy that SAACs has attracted significant research interests lately due to their immense potential in cost-effective large-scale industrial usages. Thus, while the methodological workhorse of this study, i.e., SISO with DFT inputs, has already been discussed before, the knowledge and understanding presented here are novel and suitable for Nat. Comm..
- (ii) An important difference of our work from previous similar efforts [Nature Catalysis 1, 531-539 (2018); ACS Catalysis 9, 2752-2759 (2019)] is that we have used only features that are very easy to evaluate. This makes high-throughput screening of a huge number of SAAC systems practically accessible, and we have exploited this in our study. Using our models, we have identified more than two hundred new SAACs with quantitative hierarchy for experimental validation, and have highlighted two new SAACs (Mn/Ag(111) and Pt/Zn(0001)) as particularly promising candidates. Moreover, in the updated manuscript we have also developed a novel strategy of analyzing complex models obtained by symbolic regression, based on the data-mining approach subgroup discovery (SGD).
- (iii) Besides the thermodynamic properties (i.e., binding energy, adsorption energy, and adsorption free energy) used in previous work [Nature Catalysis 1, 339-348 (2018); Nature Catalysis 1, 696-703 (2018); Nature 581, 178-183 (2020)] to describe the performance of catalysts, we have also included the kinetic property (energy barrier) and a stability indicator. As a result, our models both explain well the experimental results and enable design of high-performance catalysts with not only higher activity and but also stability.

2) While the SISO with cross validation is reasonably accurate for training a small dataset, its generalization to new systems is still the biggest problems for all current learning framework. Active learning approach was used to tackle this problem (K. Tran, Z. W. Ulissi, Active learning across intermetallics to guide discovery of electrocatalysts for CO₂ reduction and H₂ evolution. Nature Catalysis. 1, 696–703 (2018).), while a large amount of calculations are required. The current study used only ~300 datapoints for training and extend the model to ~5000 space without validation of model prediction.

Response: Indeed, the referee is correct that active learning can ensure reliability of the model. However, combination of SISO with active learning is a non-trivial task, because typical SISO model construction is computationally expensive. To address referee's concern, we analyze in more detail the cross-validation results, in particular the stability of descriptor selection during the

cross-validation. In addition, we validate our predictions by performing DFT calculations for some of the identified high-performance SAACs, including all the experimentally studied systems and our suggested top two best systems. We would also like to mention that the number of data points/systems in our training set is almost three times larger than that in the study of oxide-supported single-atom catalyst systems studied by Nolan and co-workers using the compressed-sensing LASSO approach [Nature Catalysis. 1, 531–539 (2018)].

Changes made:

We have added the sentence “For the optimal dimensionality, the same set of primary features are selected is found during CV10 in 9, 8, and 8 cases for the SISSO models of BE_H , E_b , and SE, respectively” on page 6 of the revised manuscript.

3) The criteria for screening catalysts used in this study is arbitrary. Without detailed kinetics, the approach can only provide a rough screening of candidate materials.

Response: We agree with the referee that detailed kinetics would improve reliability of the predictions. However, this is currently not feasible. Nevertheless, we do include a kinetic property (dissociation barrier), while previously only thermodynamic properties (binding energy, adsorption energy, adsorption free energy) were considered [Nature Catalysis 1, 339-348 (2018); Nature Catalysis 1, 696-703 (2018); Nature 581, 178-183 (2020)]. Thus, our study is the first to include a kinetic property in data-driven catalyst design. Alternatively, one can validate results of the predictions against experimental measurements. Our model consistently predicts high efficiency of the experimentally studied Pd/Cu, Pt/Cu, Pd/Ag, Pt/Au, Pd/Au, Pt/Ni, Au/Ru, and Ni/Zn SAACs (the first metal is the dispersed component), which further confirms its validity.

4) For segregation, a recent study by Grabow et al. (K. K. Rao, Q. K. Do, K. Pham, D. Maiti, L. C. Grabow, and Extendable Machine Learning Model for the Stability of Single Atom Alloys. Top. Catal. (2020), doi:10.1007/s11244-020-01267-2.). Even the *H binds weakly on the metals, its effect on the segregation is not considered in any of those studies.

Response: Actually, adsorption energies of H on metal surfaces are not small for some systems. For example, at room temperature and partial pressure of $H_2 = 1$ atm, free energy of adsorption for the experimentally established Pt/Ag(111) system is -0.23 eV and the H adatom induced segregation energy change is as high as 0.49 eV.

Changes made:

The following was added to the main text:

“We note that a machine-learning study of stability of single-atom metal alloys has been recently reported [Topics in Catalysis (2020) 63:728–741]. However, our analysis takes into account effects of adsorbates on the segregation energy, which has not been done previously.”

5) The most fundamental problem of this study and the approach in general is their lacking of understanding the uniqueness of single atom alloys. Although the SISSO method comes up formula in reduced feature space, the physics is missing. The message to the community by the study is rather

incremental while does not provide a way forward to tackle all those issues.

Response: We are grateful to referee for this comment, as it shows that important implications of our study were unclear. Our results show that it is exactly the uniqueness of SAACs that requires advanced data analysis techniques to predict their properties. As we demonstrate, the easy-to-understand correlations that work well for simple metal surfaces are not applicable to SAACs. We use a methodology (compressed sensing) that not only provides a model based on easily accessible features, but also identifies the level of complexity of the problem in terms of those features.

Nevertheless, we admit that additional data analysis that identifies common features of good SAACs would be useful. Therefore, we applied the subgroup discovery (SGD) approach to evaluate relative role of different features in actuating desirable changes in target properties and to facilitate physical understanding of the actuating mechanisms.

Changes made:

We have added the following paragraphs on page 10 and 11 of the revised manuscript.

“Although the SISO models are analytic formulas, the corresponding descriptors are complex, reflecting the complexity of the relationship between the primary features and the target properties. While potentially interpretable, the models do not provide a straightforward way of evaluating relative role of different features in actuating desirable changes in target properties. To facilitate physical understanding of the actuating mechanisms, we apply the subgroup discovery (SGD) approach.⁵⁵⁻⁶⁰ SGD finds local patterns in data that maximize a quality function. The patterns are described as an intersection (a selector) of simple inequalities involving provided features, e.g., (feature1<a1) AND (feature2>a2) AND... . The quality function is typically chosen such that it is maximized by subgroups balancing the number of data points in the subgroup, deviation of the median of the target property for the subgroup from the median for the whole data set, and the width of the target property distribution within the subgroup.⁶⁰”

“Here, we apply SGD in a novel context, namely as an analysis tool for symbolic regression models, including SISO. The primary features that enter the complex SISO descriptors of a given target property are used as features for SGD (see Table 2). The data set includes all 5200 materials and surfaces used in the high-throughput screening. The target properties are calculated using the obtained SISO models. Five target properties are considered: $\sqrt{\Delta G^2 + E_b^2}$, SE, SE_H, E_b, |ΔG|, and BE_H. Since we are interested mainly in catalysts that are active at normal conditions, ΔG is calculated at T = 300 K. Our goal is to find selectors that *minimize* these properties within the subgroup. Such selectors describe actuating mechanisms for minimization of a given target property. For SE, the following best selector is found: (EC* ≤ -3.85 eV) AND (-3.36 eV < EC ≤ -0.01 eV) AND (IP ≥ 7.45 eV). The corresponding subgroup contains 738 samples (14% of the whole population), and the distribution of SE within the subgroup is shown in Figure S10. Qualitatively, the first two conditions imply that the cohesive energy of the host material is larger in absolute value than the cohesive energy of the guest material. Physically this means that bonding between host atoms is preferred over bonding between guest atoms and therefore over intermediate host-guest binding. This leads to the tendency of maximizing number of host-host bonds by pushing guest atom to the surface. This stabilization mechanism has been discussed in literature,⁶¹ and here we confirm it by *data analysis*. In addition, we find that stability of SAACs requires that ionization potential of the

guest atom is high. This can be explained by the fact that lower IP results in more pronounced delocalization of the s valence electrons of the guest atom and partial charge transfer to the surrounding host atoms. The charge transfer favors larger number of neighbors due to increased Madelung potential, and therefore destabilizes surface position of the guest atom.

We calculate SE_H using SISSO models for SE and BE_H [see equation (3) in the Methods section]. Therefore, SGD for SE_H is performed using primary features appearing in the descriptors of both SE and BE_H . The top found subgroup contains features related to binding of H to the host and guest metal atoms, e.g. ($EB^* < -5.75$ eV) AND ($EH^* \leq -2.10$ eV) AND ($EH \geq -2.88$ eV) AND ($IP^* \leq 7.94$ eV) AND ($IP > 8.52$ eV) AND ($R \geq 1.29$ Å). However, the distribution of SE for this subgroup is very similar to the distribution of SE_H , which means that the stability of guest atoms at the surface is weakly affected by H adsorption when the surface guest atoms are already very stable. The important effect of H adsorption is revealed when we find subgroups minimizing directly $SE_H - SE$ (in this case only primary features that appear in the SISSO descriptor of BE_H are considered for SGD analysis). The top subgroup we found contains 72 samples (1.4% of the whole population) and is described by several degenerate selectors, in particular (-2.35 eV $\leq EH^* \leq -2.32$ eV) AND ($EC^* > -2.73$ eV) AND ($EC < -5.98$ eV) AND ($H \geq -5.12$ eV). This is a very interesting and intuitive result. Distributions of SE_H and SE for this subgroup are shown in Figure S11. The SE for all materials in the subgroup is above 0 eV. However, SE_H is much closer to 0 eV, and is below 0 eV for a significant number of materials in this subgroup. The conditions on the cohesive energy of guest and host metals (very stable bulk guest metal and less stable bulk host metal) are reversed with respect to SE, i.e., adsorption of hydrogen affects strongly the systems where guest atom is unstable at the surface. This increases the reactivity of the guest atom towards an H atom. The condition ($EH^* \geq -2.35$ eV) selects materials for which interaction of H with a host atom is not too strong, so that H can bond with the guest atom and stabilize it at the surface. The condition ($EH^* \leq -2.32$ eV) makes the subgroup narrower, which further decreases median difference $SE_H - SE$ but has no additional physical meaning. The condition ($H \geq -5.12$ eV) has a minor effect on the subgroup.

One of the top selectors (among several describing very similar data subsets) for minimizing $\sqrt{\Delta G^2 + E_b^2}$ (calculated at $T = 300$ K) is: (-2.85 eV $\leq DC \leq 1.95$ eV) AND ($DT^* \leq -0.17$ eV).

The corresponding subgroup contains 1974 samples (38% of the whole population), and the distribution of E_b within the subgroup is shown in Figure S10. The selector implies that systems providing low barrier for H_2 dissociation and at the same time balanced binding of H atoms to the surface are characterized by (i) d -band center of the bulk guest metal around the Fermi level and (ii) d -band center of the host surface top layer below the Fermi level. This can be understood as follows. Condition (i) implies that there is a significant d -electron density that can be donated to the adsorbed H_2 molecule, facilitating its dissociation. A very similar (apart from slightly different numerical values) condition appears in the selector for the best subgroup for E_b target property alone [$(-2.05$ eV $\leq DC \leq 1.46$ eV) AND ($EC^* \geq -6.33$ eV)]. Condition (ii) implies that the surface d -band center is more than half filled, which provides additional electrons for transferring to the H_2 molecule, but without excessive binding, to minimize $|\Delta G|$ in accordance with Sabatier principle. Indeed, several subgroups of strongly bound H atoms (minimizing BE_H) are described by selectors including condition $DT^* > -0.17$, which is exactly opposite to condition (ii). Analysis of BE_H and $|\Delta G|$ also

shows that the strong and intermediate binding of H atoms to the surface is fully controlled by the features of host material.

We note that SGD is capable of finding several alternative subgroups, corresponding to different mechanisms of actuating interesting changes in target properties. These subgroups have a lower quality according to the chosen quality function, but they still contain useful information about a particular mechanism. In fact, they can be rigorously defined as top subgroups under additional constraint of zero overlap (in terms of data points) with previously found top subgroups. Analysis of such subgroups can be a subject of future work. We also note that quality function used in SGD is a parameter and can affect the found subgroups. It should be chosen based on the physical context of the problem. Exploring the role of different factors in the quality function and taking into account proposition degeneracy (no or minor effect of different conditions in the selectors due to correlation between the features) allows us to develop an understanding that may not be possible without the SGD analysis.”

Reviewer 3: The authors report the use of modern data analytics towards the reliable prediction of activity and stability of dilute alloy “single atom catalysts” for hydrogenation. The topic of particular interest as single atom catalysts have made massive strides for oxidation reactions but have had limited success for reductions particularly due to lack of activity and/or abysmal stability.

1) The strength of the authors approach is that it addresses catalyst screening beyond the simple approximation BEP, d-band center etc. etc. etc. These concepts are embedded in the psyche of computational catalysis so deep that we forget they are simple models and, in many instances, too simple for quantitative predictions-but excellent for rationalizations on small data sets.

Response: We thank the reviewer for this comment. It correctly outlines the important aspect of our work.

2) The authors show that by assembling a large number of atomic, bulk and alloy descriptors (table1) they are able to perform a high dimensional correlation with the ab initio data to yield property predictions FAR more accurate than the existing simple concepts. On the one hand this is a great step forward for screening studies on the other hand if I have a more complex fitting function, I do expect a better fit. The one worry I have is this then become a brute force approach without the intellectual understanding that can be provided by a simple model. In this respect it might have been more intellectually pleasing for the authors to consider if there was a smaller subset of parameters (2-3) that might do a reasonable job (better than linear fits but not the full-blown set) which might hint at a simpler model. As is, the approach is fine I do worry about both overfitting/underfitting of data but do believe the authors have covered this ground adequately.

Response: This is a very important comment that overlaps with similar concerns of the other referees. Indeed, we perform a careful cross-validation of our models and validate them on a test set never used for training, to ensure models’ predictive power. However, the training and test sets are unavoidably limited, and there is never a guarantee that we capture all important physical variations present in the larger data set. This makes our mind crave for additional consistency check that we

call “physical understanding”. It justifies extrapolation of the models, possibly even to a different class of systems. Such extrapolation can be very useful, but also very misleading, as our study demonstrates. Nevertheless, we admit that additional data analysis that identifies common features of good SAACs would be useful. Therefore, we applied the subgroup discovery (SGD) approach to evaluate relative role of different features in actuating desirable changes in target properties and to facilitate physical understanding of the actuating mechanisms.

Changes made:

We have added the following paragraphs on page 10 and 11 of the revised manuscript.

“Although the SISSO models are analytic formulas, the corresponding descriptors are complex, reflecting the complexity of the relationship between the primary features and the target properties. While potentially interpretable, the models do not provide a straightforward way of evaluating relative role of different features in actuating desirable changes in target properties. To facilitate physical understanding of the actuating mechanisms, we apply the subgroup discovery (SGD) approach.⁵⁵⁻⁶⁰ SGD finds local patterns in data that maximize a quality function. The patterns are described as an intersection (a selector) of simple inequalities involving provided features, e.g., (feature1<a1) AND (feature2>a2) AND... . The quality function is typically chosen such that it is maximized by subgroups balancing the number of data points in the subgroup, deviation of the median of the target property for the subgroup from the median for the whole data set, and the width of the target property distribution within the subgroup.⁶⁰”

“Here, we apply SGD in a novel context, namely as an analysis tool for symbolic regression models, including SISSO. The primary features that enter the complex SISSO descriptors of a given target property are used as features for SGD (see Table 2). The data set includes all 5200 materials and surfaces used in the high-throughput screening. The target properties are calculated using the obtained SISSO models. Five target properties are considered: $\sqrt{\Delta G^2 + E_b^2}$, SE, SE_H, E_b, |ΔG|, and BE_H. Since we are interested mainly in catalysts that are active at normal conditions, ΔG is calculated at $T = 300$ K. Our goal is to find selectors that *minimize* these properties within the subgroup. Such selectors describe actuating mechanisms for minimization of a given target property. For SE, the following best selector is found: (EC* ≤ -3.85 eV) AND (-3.36 eV < EC ≤ -0.01 eV) AND (IP ≥ 7.45 eV). The corresponding subgroup contains 738 samples (14% of the whole population), and the distribution of SE within the subgroup is shown in Figure S10. Qualitatively, the first two conditions imply that the cohesive energy of the host material is larger in absolute value than the cohesive energy of the guest material. Physically this means that bonding between host atoms is preferred over bonding between guest atoms and therefore over intermediate host-guest binding. This leads to the tendency of maximizing number of host-host bonds by pushing guest atom to the surface. This stabilization mechanism has been discussed in literature,⁶¹ and here we confirm it by *data analysis*. In addition, we find that stability of SAACs requires that ionization potential of the guest atom is high. This can be explained by the fact that lower IP results in more pronounced delocalization of the *s* valence electrons of the guest atom and partial charge transfer to the surrounding host atoms. The charge transfer favors larger number of neighbors due to increased Madelung potential, and therefore destabilizes surface position of the guest atom.

We calculate SE_H using SISSO models for SE and BE_H [see equation (3) in the Methods

section]. Therefore, SGD for SE_H is performed using primary features appearing in the descriptors of both SE and BE_H . The top found subgroup contains features related to binding of H to the host and guest metal atoms, e.g. ($EB^* < -5.75$ eV) AND ($EH^* \leq -2.10$ eV) AND ($EH \geq -2.88$ eV) AND ($IP^* \leq 7.94$ eV) AND ($IP > 8.52$ eV) AND ($R \geq 1.29$ Å). However, the distribution of SE for this subgroup is very similar to the distribution of SE_H , which means that the stability of guest atoms at the surface is weakly affected by H adsorption when the surface guest atoms are already very stable. The important effect of H adsorption is revealed when we find subgroups minimizing directly $SE_H - SE$ (in this case only primary features that appear in the SISO descriptor of BE_H are considered for SGD analysis). The top subgroup we found contains 72 samples (1.4% of the whole population) and is described by several degenerate selectors, in particular (-2.35 eV $\leq EH^* \leq -2.32$ eV) AND ($EC^* > -2.73$ eV) AND ($EC < -5.98$ eV) AND ($H \geq -5.12$ eV). This is a very interesting and intuitive result. Distributions of SE_H and SE for this subgroup are shown in Figure S11. The SE for all materials in the subgroup is above 0 eV. However, SE_H is much closer to 0 eV, and is below 0 eV for a significant number of materials in this subgroup. The conditions on the cohesive energy of guest and host metals (very stable bulk guest metal and less stable bulk host metal) are reversed with respect to SE, i.e., adsorption of hydrogen affects strongly the systems where guest atom is unstable at the surface. This increases the reactivity of the guest atom towards an H atom. The condition ($EH^* \geq -2.35$ eV) selects materials for which interaction of H with a host atom is not too strong, so that H can bond with the guest atom and stabilize it at the surface. The condition ($EH^* \leq -2.32$ eV) makes the subgroup narrower, which further decreases median difference $SE_H - SE$ but has no additional physical meaning. The condition ($H \geq -5.12$ eV) has a minor effect on the subgroup.

One of the top selectors (among several describing very similar data subsets) for minimizing $\sqrt{|\Delta G|^2 + E_b^2}$ (calculated at $T = 300$ K) is: (-2.85 eV $\leq DC \leq 1.95$ eV) AND ($DT^* \leq -0.17$ eV). The corresponding subgroup contains 1974 samples (38% of the whole population), and the distribution of E_b within the subgroup is shown in Figure S10. The selector implies that systems providing low barrier for H_2 dissociation and at the same time balanced binding of H atoms to the surface are characterized by (i) d -band center of the bulk guest metal around the Fermi level and (ii) d -band center of the host surface top layer below the Fermi level. This can be understood as follows. Condition (i) implies that there is a significant d -electron density that can be donated to the adsorbed H_2 molecule, facilitating its dissociation. A very similar (apart from slightly different numerical values) condition appears in the selector for the best subgroup for E_b target property alone [$(-2.05$ eV $\leq DC \leq 1.46$ eV) AND ($EC^* \geq -6.33$ eV)]. Condition (ii) implies that the surface d -band center is more than half filled, which provides additional electrons for transferring to the H_2 molecule, but without excessive binding, to minimize $|\Delta G|$ in accordance with Sabatier principle. Indeed, several subgroups of strongly bound H atoms (minimizing BE_H) are described by selectors including condition $DT^* > -0.17$, which is exactly opposite to condition (ii). Analysis of BE_H and $|\Delta G|$ also shows that the strong and intermediate binding of H atoms to the surface is fully controlled by the features of host material.

We note that SGD is capable of finding several alternative subgroups, corresponding to different mechanisms of actuating interesting changes in target properties. These subgroups have a lower quality according to the chosen quality function, but they still contain useful information about a

particular mechanism. In fact, they can be rigorously defined as top subgroups under additional constraint of zero overlap (in terms of data points) with previously found top subgroups. Analysis of such subgroups can be a subject of future work. We also note that quality function used in SGD is a parameter and can affect the found subgroups. It should be chosen based on the physical context of the problem. Exploring the role of different factors in the quality function and taking into account proposition degeneracy (no or minor effect of different conditions in the selectors due to correlation between the features) allows us to develop an understanding that may not be possible without the SGD analysis.”

3) Finally, then the result of this study is that using their model they can rapidly predict the results of DFT calculations and use that data to make predictions about activity and stability based on simple energetic parameters such as presented in Figure 4. In my opinion this is the most important plot in the whole paper and the authors did not really deal with its ramifications very well. The wisdom in single atom catalysts (particularly for hydrogenation) is that the more active the species the less stable it will be—hence the scarcity of single atoms (dilute alloys) that are reported. If the authors are correct there is a large abundance of materials far in the lower right-hand corner (active and stable) that should break this trend whereas those that do exist are mostly in the upper right-hand corner (active but less stable). This is the most significant discovery/prediction in the paper as far as I am concerned, and the authors barely comment on it. Sadly, a follow-on experimental study making targets and validating the prediction would be a breakthrough and this is also not done.

Response: There seems to be a misunderstanding regarding Fig. 4. The most active and stable materials are in the lower LEFT-hand corner. Just as the referee points out, this corner is scarcely populated compared to the whole area covered by all calculated materials. However, this does not mean there are no materials that can be better than the experimentally tested ones. To clarify this aspect, we have now added a discussion to the main text and new Figure S9 in the revised supporting information, which is reproduced as Figure R1 below.

Figure R1. Stability vs. activity map for flat SAACs surfaces at T=298 K and p=1 atm. The SE on y-axis represents stability and activity parameter $\sqrt{\Delta G^2 + E_b^2}$ is shown on x-axis.

Changes made:

1) We have added the sentences:

“As expected, stability and activity are inversely related, which can be seen from the negative slope of the general trend in Figure 8 (showing selected materials) and Figure S9 (showing all explored materials), as well as a cut-off in population of the lower left-hand corner of these plots. Nevertheless, there are several materials that are predicted to be better SAACs than the so-far reported ones.” on page 11 of the revised manuscript.

2) We added Figure S9 in the revised supporting materials.

4) Sadly, a follow-on experimental study making targets and validating the prediction would be a breakthrough and this is also not done.

Response: This work was conceived as a theoretical one. We are happy to share methodology and predictions with the community as soon as possible. We very much hope that our findings will encourage experimental groups to validate our predictions.

REVIEWERS' COMMENTS

Reviewer #1 (Remarks to the Author):

The authors have greatly expanded their work based on the reviewer comments. Importantly, they now utilize a data mining algorithm called Subgroup Discovery to analyze their SAAC dataset in combination with their SISSO model. This added analysis enables the authors to give much more satisfying and general insights regarding the stability and activity of the SAACs, which should prove useful for the catalysis community. Additionally, Subgroup Discovery has not been used yet in the catalysis/surface science fields (and SISSO algorithm has only been used once before in catalysis field to my knowledge), thus this work also introduces cutting-edge data science tools to the broader scientific community. Therefore, this paper should be of broad interest to multiple communities. I believe the work is suitable for publication.

Reviewer #2 (Remarks to the Author):

Authors addressed most of the comments. However, the physical insights by subgroup discovery is rather limited. I stick to my opinion that this work is not a significant step toward ML method itself or SAAC discovery. It might be appropriate to a more specialized catalysis journal.

1. The SISSO machine learning method employed in this study is not new. With the same set of features, a regular neural network can be more easily trained and coupled with active learning. With existing alloy database published in community, a convolutional neural net can also be used since the local environment of single atom alloys is analogous to the traditional fcc-type alloys, e.g. A3B, in the first coordination shell. In term of physical interpretation, they are all black-box models. SISSO can give a formula instead, although its direct understanding by a catalysis expert is still not there. The formula can be considered as symbolic regression rather than physical models. Interpreting black-box models are not necessarily providing physical insights that can be translated to design.

2. Subgroup discovery is a half-way approach to extract conditions of features optimizing a defined quality function. It is monte carlo based algorithm. The identified boundary values will depend on runs and hyperparameters. The approach has been used in materials science and catalysis. It is overstated in terms of novelty in the context. The rule identified by the method is convoluted rather than being insightful.

3. The design space of SAAs is relatively small compared to complex alloys. The indication in abstract for hundreds of thousands is misleading.

4. It says the energy BEH and the d-band center and (b) the H₂ dissociation energy barrier E_b and the H₂ dissociation reaction energy for Pt(111) based SAACs. But the (b) panel is missing.

5. It claimed a step away from the d-band theory, BEP, and scaling relations. While machine learning models can be considered as a further step away from the d-band center type of theory level, it is not fair to say that for the original d-band theory since machine learning models are regression based only. It is not close to go beyond BEP and scaling relations in this work since it simply does not consider full reaction pathways. The claim is irrelevant.

6. The d-band center of the bonding guest atom is obvious choice for atop adsorption, but not quite for hollow, bridge. The averaged d-band center of a collection of atoms in the revision is not the right since the coupling strength decays rapidly with distance.

Reviewer #3 (Remarks to the Author):

After carefully considering the previous reviewers' comments and the revised manuscript I can say most of the technical concerns I have about this work are resolved and I may have even softened (but not changed) my stance about not really bringing new understanding. I still do not like these screening/data analytics papers for the sake of data analytics but in this case the decision point for me is that the authors predict many new catalysts so, in principle, the way to test and validate this model is on the table.

IF the authors are right then this is a breakthrough, if they are wrong ... I think this may well be worth publishing in Nature Comm and I look forward to seeing this work validated (or not).

The text does require significant proof reading and improving on the English, particularly the new parts and should be proof read careful before it is published.

Reviewer 1:

The authors have greatly expanded their work based on the reviewer comments. Importantly, they now utilize a data mining algorithm called Subgroup Discovery to analyze their SAAC dataset in combination with their SISO model. This added analysis enables the authors to give much more satisfying and general insights regarding the stability and activity of the SAACs, which should prove useful for the catalysis community. Additionally, Subgroup Discovery has not been used yet in the catalysis/surface science fields (and SISO algorithm has only been used once before in catalysis field to my knowledge), thus this work also introduces cutting-edge data science tools to the broader scientific community. Therefore, this paper should be of broad interest to multiple communities. I believe the work is suitable for publication.

Reply: We thank the reviewer for the positive assessment of our work, and most of all for the reviewer's critical comments that helped us to improve our manuscript.

Reviewer 2:

Authors addressed most of the comments. However, the physical insights by subgroup discovery is rather limited. I stick to my opinion that this work is not a significant step toward ML method itself or SAAC discovery. It might be appropriate to a more specialized catalysis journal.

Reply: We thank the reviewer for sharing his/her opinion, and address the reviewer's concerns below.

1. The SISO machine learning method employed in this study is not new. With the same set of features, a regular neural network can be more easily trained and coupled with active learning. With existing alloy database published in community, a convolutional neural net can also be used since the local environment of single atom alloys is analogous to the traditional fcc-type alloys, e.g. A3B, in the first coordination shell.

In term of physical interpretation, they are all black-box models. SISO can give a formula instead, although its direct understanding by a catalysis expert is still not there. The formula can be considered as symbolic regression rather than physical models. Interpreting black-box models are not necessarily providing physical insights that can be translated to design.

Reply: Whether it would be easier to train a convolutional neural network coupled with active learning remains to be seen, in particular if one takes into account typically much larger number of training data needed for a reliable NN model. Also, it remains to be seen whether the list of primary features plus the existing data on fcc-type alloys are sufficient for a better prediction model. These questions are an interesting topic for future work, as there is no evidence in the literature regarding these issues. However, they are beyond the scope of this work. Compressed-sensing based descriptor identification is not a new idea in general, but it is much newer than NN, and we are aware of only one published application of SISO in catalysis (with a contribution by one of the authors), which however did not include a high-throughput study. Using SISO to specifically predict properties of SAACs is in fact a new idea. As the referee points out, SISO provides analytic formulas as models. While it is indeed sometimes not easy to interpret these models, in our manuscript we developed a

novel approach that allows us to move forward also in this direction. We were inspired by comments of this and the other referees to do this. Our approach allows us to extract physical understanding based on thorough data analysis rather than intuition based on a limited set of data. We believe this is an important step that is of general interest to the community, in addition to our actual high-throughput predictions.

2. Subgroup discovery is a half-way approach to extract conditions of features optimizing a defined quality function. It is monte carlo based algorithm. The identified boundary values will depend on runs and hyperparameters. The approach has been used in materials science and catalysis. It is overstated in terms of novelty in the context. The rule identified by the method is convoluted rather than being insightful.

Reply: We are not sure what referee means by “half-way approach”. We have not seen such a characteristics in existing literature. It is for sure a novel approach in catalysis. Although there is a publication that employed SGD in the context of catalysis, there are no examples of using SGD for catalyst design. However, we do not even claim novelty of SGD, we claim novelty of how we use it: to interpret complex symbolic regression models.

Yes, SGD results depend on the run if the Monte Carlo approach is used (as in our case). However, this dependence comes mainly from feature correlations, provided an extensive sampling was performed. The realkd implementation takes special care in distributing sampling points optimally to improve sampling efficiency, and we performed an extensive sampling. In fact, we always see consistency among several top subgroups. The problem is not the convolution but the idea that one can always just take an arbitrary set of features, look at the top subgroup for a given property and hope to get an insight just looking at it. As we describe in the text, the insight should come from understanding the physics behind the primary features and analysis of several top selectors and a joint analysis of subgroups for different target properties. To emphasize this, we have updated the following paragraph:

“Exploring the role of different factors in the quality function and taking into account proposition degeneracy (no or minor effect of different conditions in the selectors due to correlation between the features) can significantly improve interpretability of the selectors. The interpretability also depends crucially on our physical understanding of the features and relations between them. Nevertheless, in combination with human knowledge SGD analysis allows for development of understanding that would not be possible without the help of artificial intelligence.”

3. The design space of SAAs is relatively small compared to complex alloys. The indication in abstract for hundreds of thousands is misleading.

Reply: For binary alloys we agree with the referee, but if we consider multi-component alloys there are easily hundreds of thousands. However, since we consider only binary SAACs, we follow referee’s suggestion and remove that statement.

4. It says the energy BEH and the d-band center and (b) the H2 dissociation energy barrier E_b and the H2 dissociation reaction energy for Pt(111) based SAACs. But the (b) panel is missing.

Reply: Thank you for noticing this. Perhaps something went wrong with formatting. We have checked to make the changes correspondingly.

5. It claimed a step away from the d-band theory, BEP, and scaling relations. While machine learning models can be considered as a further step away from the d-band center type of theory level, it is not fair to say that for the original d-band theory since machine learning models are regression based only. It is not close to go beyond BEP and scaling relations in this work since it simply does not consider full reaction pathways. The claim is irrelevant.

Reply: Indeed, for a neural network model based on hand-picked descriptors this would be a fair criticism. But we use SISO to IDENTIFY descriptors, and we can directly compare their performance with the d-band center descriptor. Whether we can provide a physical interpretation of the identified descriptor is a different matter, and we show how to do it with SGD.

6. The d-band center of the bonding guest atom is obvious choice for atop adsorption, but not quite for hollow, bridge. The averaged d-band center of a collection of atoms in the revision is not the right since the coupling strength decays rapidly with distance.

Reply: For all Ag(110) and Pt(111) based SAACs the most stable adsorption sites are hollow sites. We found the correlation between BE_H and the d-band center of the d orbitals that are projected to the single guest atom for the alloyed systems provides better correlation with other properties than d-band centers for the d orbitals projected on (i) the single guest atom plus its 1st nearest neighbor shell or (ii) the whole slab. This is consistent with a previous study [Topics in Catalysis 61, 462-474 (2018)].

Reviewer 3:

After carefully considering the previous reviewers' comments and the revised manuscript I can say most of the technical concerns I have about this work are resolved and I may have even softened (but not changed) my stance about not really bringing new understanding. I still do not like these screening/data analytics papers for the sake of data analytics but in this case the decision point for me is that the authors predict many new catalysts so, in principle, the way to test and validate this model is on the table.

IF the authors are right then this is a breakthrough, if they are wrong ... I think this may well be worth publishing in Nature Comm and I look forward to seeing this work validated (or not).

The text does require significant proof reading and improving on the English, particularly the new parts and should be proof read careful before it is published.

Reply: We thank the referee for the clear opinion and critical comments that helped to significantly improve our manuscript. Indeed, to the best of our knowledge this work is the first one that uses SISO for high-throughput PREDICTIONS of catalytic properties rather than just data analysis. From this point of view, our work is both a guide for a rational design of SAACs and an important step towards testing and further development of data-analytics methodology for catalysis.

Following the advice by the referee, we have carefully proofread the text.